# Aligning LLMs by Predicting Preferences from User Writing Samples

**Stéphane Aroca-Ouellette** [1 2]   **Natalie Mackraz** [3]   **Barry-John Theobald** [3]   **Katherine Metcalf** [3]

## Abstract

Accommodating human preferences is essential for creating aligned LLM agents that deliver personalized and effective interactions. Recent work has shown the potential for LLMs acting as writing agents to infer a description of user preferences. Agent alignment then comes from conditioning on the inferred preference description. However, existing methods often produce generic preference descriptions that fail to capture the unique and individualized nature of human preferences. This paper introduces PROSE, a method designed to enhance the precision of preference descriptions inferred from user writing samples. PROSE incorporates two key elements: (1) iterative refinement of inferred preferences, and (2) verification of inferred preferences across multiple user writing samples. We evaluate PROSE with several LLMs (i.e., Qwen2.5 7B and 72B Instruct, GPT-mini, and GPT-4o) on a summarization and an email writing task. We find that PROSE more accurately infers nuanced human preferences, improving the quality of the writing agent's generations over CIPHER (a state-of-the-art method for inferring preferences) by 33%. Lastly, we demonstrate that ICL and PROSE are complementary methods, and combining them provides up to a 9% improvement over ICL alone. Code: `https://github.com/apple/ml-predict`.

## 1. Introduction

People increasingly rely on LLM-powered AI Assistants to complete tasks on their behalf, such as creating written materials: "write a professional email about the following great idea" or "summarize this new article for me to share share with my friends". As the writing style learned by an LLM during pretraining is generic, it may not match the user's preferred writing style and voice (Chakrabarty et al., 2024; Santurkar et al., 2023), leading to outputs that feel impersonal, misaligned, or requiring extensive editing.

Existing approaches to learn preferences rely on preference rankings (RLHF) (Ziegler et al., 2019; Rafailov et al., 2024), demonstrations (Ouyang et al., 2022; Shaikh et al., 2024), prompting (Zhou et al., 2022), and user edits (Gao et al., 2024). However, methods such as RLHF and SFT (on user demonstrations) require a large number of samples, and do not learn the preferences in a form users can interpret or interact with. In-context learning (ICL) from user demonstrations does learn from a small number of user demonstrations, but lacks interpretability and offers limited control to the user, and prompting approaches require the challenging task of identifying a high-quality prompt (Zamfirescu-Pereira et al., 2023). Furthermore, methods that learn from user edits ignore data about user preferences and style that are available from observing how the user completes writing tasks on their own.

Gao et al. (2024) introduces CIPHER to establish the benefits of aligning a LLM through prompting by learning a description of user preferences compared to ICL conditioned on user demonstrations (i.e., needing fewer tokens, interpretable representation, and a modifiable representation). The preference description is learned from user edits on the assistant's generations. However, CIPHER does not enable the LLM to reflect on and refine its inferred preference description, which limits the assistant's ability to adapt to a user nuanced writing style.

In this paper, we build on CIPHER and introduce **PROSE** (**P**reference **R**easoning by **O**bserving and **S**ynthesizing **E**xamples), a novel approach that leverages two key innovations to enhance the precision and efficacy of the preference description inferred from user demonstrations: (1) iteratively refining the inferred description until the assistant's generations closely align with the user, and (2) verifying the inferred preferences across multiple user demonstrations. The inferred description is used to condition the LLM to generate writing more aligned with the user.

We evaluate PROSE on PRELUDE (Gao et al., 2024), the

---

[1] Work done during internship. [2] Department of Computer Science, University of Colorado, Boulder, CO USA [3] Apple, Cupertino, CA USA. Correspondence to: Stéphane Aroca-Ouellette <stephane.aroca-ouellette@colorado.edu>, Katherine Metcalf <kmetcalf@apple.com>.

*Proceedings of the 42nd International Conference on Machine Learning*, Vancouver, Canada. PMLR 267, 2025. Copyright 2025 by the author(s).

assistive writing benchmark accompanying CIPHER, and identify several limitations. First, the ground truth preference sets often overlap, lack diversity, and match the default LLM behavior. Second, edits are performed only if the assistant generation is inadequate, meaning it is not possible to distinguish between good and excellent generations. Lastly, PRELUDE relies on user edits as the learning signal, meaning the assistant's initial draft can limit the quality of the final writing sample. To address these limitations, we introduce a novel assistive writing benchmark, PLUME (**P**reference **L**earning from **U**ser **E**mails and **M**emos).

We systematically evaluate the benefits of PROSE on PLUME using four LLMs ranging in size and ability, and find that PROSE outperforms CIPHER by 33% (Gao et al., 2024). Additionally, we demonstrate PROSE can be combined with ICL to further improve over CIPHER by 47% and up to 9% over ICL. In all, our contributions are:

- PROSE: A new method to infer user preferences.
- PLUME: An improved benchmark for preference inference from user writing demonstrations.
- An in-depth ablation study on PROSE's iterative refinement and consistency verification steps
- An analysis comparing learning explicit preference descriptions and conditioning directly on in-context examples

## 2. Related Work

**Personalizing LLMs** In natural language generation, prompting (Radford et al., 2019) and in-context learning (Brown et al., 2020) have proven effective methods for controlling the generation of text, especially in a preference-driven context (Sun et al., 2023; 2024).

Some prior approaches for adapting models to user preferences involve RLHF (Stiennon et al., 2020) and fine-tuning (Tan et al., 2024; Zhuang et al., 2024), which can be compute-intensive and inaccessible to some practitioners without the budget or scale of needed data. To reduce data requirements, Shaikh et al. (2024) propose treating user demonstrations as implicitly preferred over all model outputs, allowing for more efficient preference modeling. Another line of work aims to minimize compute demands by identifying and selectively adjusting internal activations to steer model behavior (Li et al., 2023; Turner et al., 2024; Lindsey et al., 2025). While effective for promoting broad, predefined objectives—such as improving truthfulness or reducing toxicity—it remains unclear how such techniques can generalize to individual users without explicit guidance. With the rise of LLMs with strong instruction-following capabilities, methods like prompting to adapt to a user's profile have become more popular (Shen et al., 2024; Salemi et al., 2024); however, these methods too often rely on explicit user feedback to optimize prompts (Lin et al., 2024). PROSE

circumvents these issues by learning from implicit user signals, breaking down preferences into sub-components to generate tailored user-preferences, all without the need for fine-tuning.

**Preference-Conditioned Agents** Combined preference inference and conditioning has recently gained traction, with the following three works most aligned with PROSE.

Peng et al. (2024) explores preference learning in quadrupedal mobile manipulation using an object detection module to map image observations to text. An LLM then infers preferences by comparing pairs of trajectories. These preferences are in turn used to improve task alignment with user preferences. Shashidhar et al. (2024) train a preference inferring model that outputs a set of rules to use during generation, and demonstrate improved personalization on a set of writing tasks. Lastly, Gao et al. (2024) propose the PRELUDE environment, where an LLM learns writing style preferences in a collaborative authoring task. We discuss this work in detail in Section 4.

These methods all rely on a single inference step, whereas our approach uses iterative refinement to learn more precise preferences, and preference verification across several user examples for robustness.

## 3. PROSE

PROSE aligns an AI writing assistant with a user's preferences $\bar{\mathbf{p}}_u$ by learning a preference description $\hat{p}_{\text{desc}}$ that allows the assistant (an LLM) to mimic the user's demonstrations $\mathbf{w}_u$, which are determined by $\bar{\mathbf{p}}_u$. For example, learning that articles should be summarized in the style of an old timey radio broadcast.

Each time the user gives the assistant a new task or provides a new task-description and demonstration pair $(x_{\text{task}}, w_u)$, following (Gao et al., 2024) PROSE retrieves up to three previously observed demonstrations relevant to the given task along with the preferences inferred from those demonstrations from its interaction memory. The retrieved preferences are then aggregated to form the preference description $\hat{p}_{\text{desc}}$ using the prompt in Figure 8 (Appendix F.1), which is used to condition the assistant during generation: $w_a = \texttt{generate}(\texttt{llm}, x_{\text{task}}, \hat{p}_{\text{desc}})$. If no demonstrations have been seen, the AI assistant is not conditioned on any preferences, $w_a = \texttt{generate}(\texttt{llm}, x_{\text{task}})$.

If the AI assistant's generation, $w_a$ does not match the user's demonstration, $w_u$, the inferred preference description $\hat{p}_{\text{desc}}^0$ is updated via **iterative refinement** (Section 3.1) steps and a **preference consistency verification** (Section 3.2) step – PROSE's contributions. Iterative refinement alternates between updating the inferred preference description $\hat{p}_{\text{desc}}^{s+1}$ by comparing the agent's generation, $w_a$, to the user's demon-

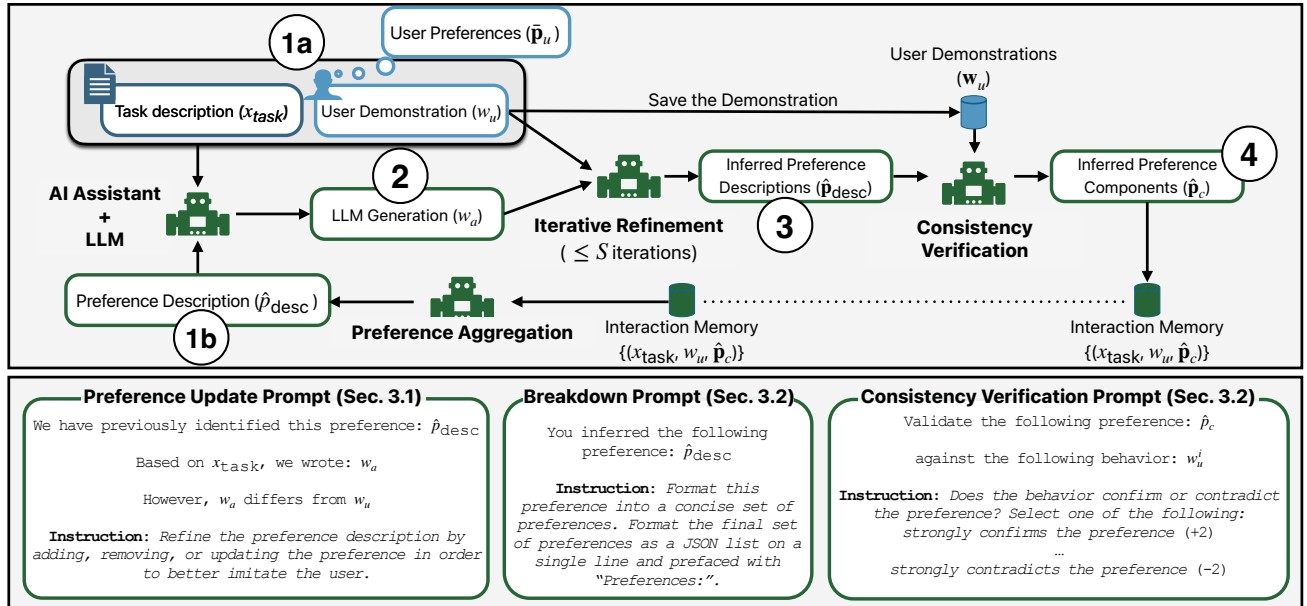

Figure 1: **Overview of PROSE**. (top) The user provides a task description and demonstration to PROSE, which executes iterative refinement and then a consistency verification step. Iterative refinement updates the inferred preference description by generating a writing sample conditioned on the current preference description, comparing the sample to the user's demonstration, and updating the preference description to better describe the user's demonstration until the LLM's generations match the demonstration or a maximum number of iterations $S$ is reached. The description is then broken into a set of component parts, and each component's consistency with prior demonstrations is verified with LLM-as-a-Judge. (bottom) Example PROSE prompts (for full prompts see Appendix F.1).

stration, $w_u$, and rerunning generation conditioned on the updated $\hat{p}_{\text{desc}}^{s+1}$ until either the maximum number of iterative refinement steps ($S$) is reached or no updates to the inferred preference description are made. Consistency verification breaks the final $\hat{p}_{\text{desc}}$ into preference components and prunes components that are not supported by previously seen demonstrations.

A visualization of PROSE (top) and the prompt summaries (bottom) for each of its preference inference steps are provided in Figure 1. The algorithm is provided in Appendix A, and the complete prompts are in Figure 8 (Appendix F.1)[1].

### 3.1. Iterative Refinement

To improve $\hat{p}_{\text{desc}}$, the LLM is prompted to compare and contrast $w_a$ and $w_u$ and then modify $\hat{p}_{\text{desc}}$ such that the modification reduces the difference between $w_a$ and $w_u$: $\hat{p}_{\text{desc}}^{s+1} = \texttt{generate}(\texttt{llm}, x_{\text{update}}, \hat{p}_{\text{desc}}^s, w_u, w_a)$, where $x_{\text{update}} =$"Preference Update Prompt" in Figure 1. The updated preference description is accumulated in $\hat{\mathbf{p}}_{\text{desc}} = [\hat{p}_{\text{desc}}^0, ..., \hat{p}_{\text{desc}}^s]$, where $s$ is the iterative refinement step.

PROSE then conditions the LLM on the updated preference description $\hat{p}_{\text{desc}}^{s+1}$ to generate a new writing sample $w_a^{s+1}$.

The process of generating AI assistant writing samples, comparing to the user demonstrations, and updating the inferred preferences continues until either the candidate solutions exactly match the user's demonstrations, the preference description is unchanged between subsequent update steps, or a maximum number of iteration steps is reached ($S$). Qualitative examples of the consistency verification procedure are in Appendix F.5.

### 3.2. Consistency Verification

After the preference description is improved through iterative refinement, each component of each preference description in $\hat{\mathbf{p}}_{\text{desc}}$ is verified against relevant, previously observed user demonstrations. The verification step removes preference components that were incorrectly inferred or are overly specific to a single demonstration.

Consistency verification operates on the component level (e.g. "use emojis", "use alliterations"). Therefore, the natural language preference descriptions (e.g. "write a tweet with emojis and alliterations") produced by iterative refinement are first broken into components by prompting the LLM to convert the preference description into an ordered set of preference components. The preference components are aggregated over all preference descriptions to help avoid over

[1] code coming soon!

fitting: $\hat{\mathbf{p}}_c = \bigcup_{s=0}^{|\hat{\mathbf{p}}_{\text{desc}}|}(\texttt{generate}(\texttt{llm}, x_{\text{breakdown}}, \hat{p}^s_{\text{desc}}))$, where $x_{\text{breakdown}} =$"Breakdown Prompt" in Figure 1.

PROSE verifies each preference component in $\hat{\mathbf{p}}_c$ against each of the relevant user demonstrations by prompting an LLM to assign a score $v^i_{\text{score}} \in [-2, 2]$ indicating how strongly the demonstration confirms the preference: $v_{\text{score}} = \frac{1}{|\mathbf{w}_u|} \sum_{i=0}^{|\mathbf{w}_u|}(\texttt{generate}(\texttt{llm}, x_{\text{verification}}, \mathbf{w}^i_u, \hat{p}^s_{\text{desc}}))$, where $x_{\text{verification}}$ is "Consistency Verification Prompt" in Figure 1. If $v_{\text{score}}$ is below the specified threshold ($v$), the preference component is removed. The task description, user demonstration, and final preference components $(x_{\text{task}}, w_u, \hat{\mathbf{p}}_c)$ are then stored in PROSE's interactive memory. Qualitative examples of the consistency verification procedure are in Appendix F.4.

## 4. Assistive Writing Benchmark

### 4.1. PRELUDE

Gao et al. (2024) propose PRELUDE (**PRE**ference **L**earning from **U**ser's **D**irect **E**dits) to evaluate algorithms that infer preferences for assistive writing tasks. Success is defined as: (1) maximizing the quality of the inferred user's preferences and (2) minimizing the amount of work required by a user to edit the generated text into an acceptable form.

PRELUDE consists of two tasks: summarizing articles and writing emails from notes. Each task has a set of users, and each user has distinct preferences per summary and email topic (e.g., summarize an encyclopedia article versus news article). The summarization and email writing tasks have five and four users respectively. See Table 9 ( Appendix D) for the mapping between users, topics, and preferences.

To solve a given task, the agent must write a summary or email using the provided article or notes along with any preferences the agent has inferred. The user is then asked if the agent's generation is satisfactory based on the user's true preference. If the agent's generation is satisfactory, the agent accrues no penalty. If the agent's generation is not satisfactory, the user edits the agent's generation, and the agent is penalized based on the extent of the edits. The agent observes the user's edits to improve its inferred preferences.

We analyze PRELUDE and find that the (1) chosen metrics, (2) the editing process, and (3) the ground truth preferences are key limitations of the benchmark, that lead to a weak correlation between the quality of the inferred preferences and the quality of the generated writing.

**Metric Correlation** As the goal is to infer user preferences, the measure of the agent's generation quality (i.e., the user-edit-based penalty) must be highly correlated the quality of inferred preference. We measure the correlation between PRELUDE's *preference quality metric* — preference ac-

| Metric | PRELUDE | | PLUME | |
|---|---|---|---|---|
| | Acc. | P. Sim. | Acc. | P. Sim. |
| Levenshtein dist | -0.43 | -0.39 | 0.01 | -0.11 |
| PPCM | 0.42 | **0.42** | 0.39 | **0.73** |

Table 1: Subset of Pearson correlation ($\rho_{P,G}$) between preference quality metrics and generation quality metrics across both the summarization and email tasks. Best correlation in each framework is bold. P. Sim. (Preference similarity) and PPCM (Per Preference-Component Match) are described in Section 4.2. Full results in Appendix C.1.

curacy[2] — and *generation quality metric* — Levenshtein distance (Levenshtein, 1966) between the LLM generation and user edited generation. For a each summary and email topic, we generate the powerset of PRELUDE's ground truth preferences and create a population of agents. Each agent is conditioned on a subset from the powerset and completes its assigned task for five seeds. The quality of the inferred preferences and of the resulting generations is measured according to PRELUDE's performance metrics. We calculate the Pearson correlation between each of PRELUDE's *preference quality* and *generation quality* metrics:

$$\rho_{P,G} = \frac{\text{Cov}(P,G)}{\sigma_P \sigma_G}$$

where P denotes the measured preference quality and G denotes the measured generation quality. We report a subset of the results in Table 1 (Full results in  Appendix C.1).

The results, reported in Table 1, show a weak correlation ($< 0.5$) between PRELUDE's preference accuracy and Levenshtein distance metrics. The accuracy metric relies on the "highest" BERTScore, and therefore cannot differentiate partially correct preferences from perfectly correct preferences. Moreover, the Levenshtein distance varies substantially between generations even when conditioned on the exact same preferences (an illustrative example is in Appendix E.1). Gao et al. (2024) allude to this as a motivation for their two-stage editing process, and when we compare the results to a version of PRELUDE where the user always generates summaries or emails directly from the article or notes instead of editing the agent's summary or email (PRELUDE$_{\text{NoEdit}}$), we see a further drop in correlation. However, we propose addressing this issue using improved metrics.

**The Editing Procedure** Relying on a binary label to indicate whether a generation matches the user's preferences is inherently ambiguous. It is not possible to distinguish between generations that align with 65% versus 100% of preferences. Even if this ambiguity is resolved, genera-

---

[2]a preference is correct if its BERTScore (Zhang* et al., 2020) with true preference set is greater than the BERTScore with any other preference set.

tions not selected for editing incur no cost and provide no incentive to further improve the quality of the inferred preferences. Lastly, the editing process unduly influences the user's writing, as demonstrated in Appendix E.2.

**Preference Sets** We observe the following limitations with PRELUDE's preference sets: (1) certain preference components have minimal impact on the generated text, due to unclear definitions (e.g., "skillful foreshadowing") or similarity to default LLM behavior (e.g., "clear"); (2) preferences are repeated across several task topics (e.g., "short", "brief", "concise" appear in four of five summarization preference sets); and (3) there is a large variance in preference set complexities across users (e.g., "targeted to young children, storytelling, short sentences, playful language, interactive, positive" vs."question answering style"). PRELUDE's preferences are in Appendix D (Table 9)

**Knowledge of Topics** Instead of treating each task topic as a distinct user, PRELUDE introduces the additional challenge of context awareness; each user has different preferences based on the task's topic. Therefore, prior to writing a summary or an email the agent must first identify the correct context, an orthogonal challenge to inferring preferences.

### 4.2. PLUME

To address PRELUDE's limitations, we develop a new environment PLUME (**P**reference **L**earning from **U**ser **M**emos and **E**mails) based on same underlying tasks and topics as PRELUDE. As in (Gao et al., 2024), PLUME uses GPT-4o as a proxy human user. In the following sections, we provide a detailed description of how PLUME addresses each of PRELUDE's limitation.

**Metric Correlation** We investigate and compare new preference and generation-quality metrics. For the *preference quality metric*, we evaluate an LLM-as-a-Judge (Zheng et al., 2023) metric that prompts an LLM to identify how similar the inferred preference description is to the true preference description on a 5-point Likert scale, which we call Preference-Similarity. For the *generation quality metric*, we evaluate length-normalized Levenshtein distance (ln-Ldist), BERTScore, and an LLM-as-a-Judge (Zheng et al., 2023) metric inspired from the editing procedure in PRELUDE. The LLM-as-a-Judge evaluation is a per preference-component match (PPCM) that asks an LLM how much a component of a the ground truth preference is exhibited in a piece of writing on a five point Likert scale from "clearly contradicts" (score of -2) to "clearly exhibits" (score of +2). This is repeated for each component of the true preference set, and we compute the mean score across components. The full prompts used for both of the LLM-as-a-Judge metrics are shown in Appendix B (Figure 4 and Figure 5).

The results in Table 3 (Appendix C.1) show that Preference-

Similarity has a stronger correlation with each writing generation metric than PRELUDE's accuracy metric. Looking at the generation quality metrics, Levenshtein distance consistently has the weakest correlation and PPCM the strongest. Notably, the pairing of Preference-Similarity (preference quality) and PPCM (generation quality) provides the highest correlation in every situation and are the primary metrics we report in PLUME.

**The Editing Procedure** In place of the editing, PLUME has the agent and user independently solve each task to (1) enable the agent to learn from every user example, unless the agent's generation exactly matches the user's; (2) remove ambiguity about whether a generation should be edited and incur a cost; (3) provides a smoother curve along which to evaluate different methods; and (4) prevents agents from influencing users.

**Preference Sets** PLUME reworks the preferences according to the following criteria: (1) each preference set contains an equal number of components; (2) within each task, preference sets have a shared structure; (3) as much as possible, preferences components are orthogonal to each other, avoiding overlapping preferences (e.g., "write in the style of old-timey radio" and "use archaic language") or contradictory preferences (e.g., "use emojis" and "use a formal tone"); and (4) preferences components do not follow the LLMs default behavior — i.e., generating an output conditioned on no preference should lead to a lower score than when generating on the preference component. PLUME's preferences are in Appendix D (Table 9). We encourage future researchers to use PLUME with different preference sets to adjust difficulty or examine specific concepts.

**Knowledge of Topics** As this work focuses on how to infer preferences, the version of PLUME used in all experiments assumes a distinct known user per topic. We note that PLUME is easily adaptable to use hidden topics if desired.

## 5. Experimental Set Up

All experiments consist of three phases. First, the user provides a demonstration using their true preferences. Second, the agent completes the user's task using its currently inferred preferences (if any). Finally, the agent compares its generation with the user's example to infer new preferences to use going forward.

All AI assistants are evaluated on their ability to complete email writing and article summarization tasks on behalf of the user. Each task has different types (e.g., email to your boss versus email to a family member), and each user's preferences differ based on the task type. The assistants are evaluated along two dimensions: *preference quality* to measure the similarity between true and inferred preferences (see Appendix B.1), and *generation quality* to evaluate how

| Method | Summarization | | Emails | | Tasks Mean | |
|---|---|---|---|---|---|---|
| | Pref. Sim. | PPCM | Pref. Sim. | PPCM | Pref. Sim. | PPCM |
| No Learning Baselines | | | | | | |
| NPC | $0.00_{\pm 0.00}$ | $-1.09_{\pm 0.03}$ | $0.00_{\pm 0.00}$ | $-0.91_{\pm 0.03}$ | $0.00_{\pm 0.00}$ | $-1.00_{\pm 0.02}$ |
| Oracle | $3.86_{\pm 0.07}$ | $1.71_{\pm 0.04}$ | $3.89_{\pm 0.06}$ | $1.95_{\pm 0.01}$ | $3.87_{\pm 0.05}$ | $1.83_{\pm 0.02}$ |
| Learning Baselines | | | | | | |
| ICL | $0.00_{\pm 0.00}$ | $\mathbf{1.35_{\pm 0.08}}$ | $0.00_{\pm 0.00}$ | $1.39_{\pm 0.07}$ | $0.00_{\pm 0.00}$ | $1.37_{\pm 0.05}$ |
| CIPHER-1 | $1.21_{\pm 0.04}$ | $-0.05_{\pm 0.06}$ | $\underline{1.67_{\pm 0.07}}$ | $0.33_{\pm 0.05}$ | $1.44_{\pm 0.04}$ | $0.14_{\pm 0.04}$ |
| CIPHER-5 | $1.24_{\pm 0.07}$ | $-0.08_{\pm 0.09}$ | $\mathbf{1.69_{\pm 0.07}}$ | $0.25_{\pm 0.07}$ | $\underline{1.46_{\pm 0.05}}$ | $0.09_{\pm 0.06}$ |
| PROSE Ablations | | | | | | |
| $\text{PROSE}_{CE}$ | $1.23_{\pm 0.06}$ | $0.51_{\pm 0.08}$ | $1.46_{\pm 0.07}$ | $0.97_{\pm 0.08}$ | $1.34_{\pm 0.05}$ | $0.74_{\pm 0.06}$ |
| $\text{PROSE}_{u}$ | $1.30_{\pm 0.11}$ | $0.47_{\pm 0.10}$ | $1.34_{\pm 0.10}$ | $0.84_{\pm 0.11}$ | $1.32_{\pm 0.07}$ | $0.65_{\pm 0.07}$ |
| $\text{PROSE}_{u,a}$ | $1.35_{\pm 0.10}$ | $0.49_{\pm 0.11}$ | $1.58_{\pm 0.09}$ | $1.04_{\pm 0.06}$ | $1.47_{\pm 0.07}$ | $0.76_{\pm 0.06}$ |
| $\text{PROSE}_{u,a,S>1}$ | $1.37_{\pm 0.11}$ | $0.75_{\pm 0.09}$ | $1.50_{\pm 0.08}$ | $1.21_{\pm 0.08}$ | $1.43_{\pm 0.07}$ | $0.98_{\pm 0.06}$ |
| $\text{PROSE}_{NV}$ | $1.47_{\pm 0.06}$ | $0.87_{\pm 0.10}$ | $1.38_{\pm 0.10}$ | $1.18_{\pm 0.08}$ | $1.43_{\pm 0.06}$ | $1.02_{\pm 0.06}$ |
| $\text{PROSE}_{Full}$ | $\mathbf{1.51_{\pm 0.09}}$ | $0.90_{\pm 0.07}$ | $1.47_{\pm 0.08}$ | $1.24_{\pm 0.07}$ | $\mathbf{1.49_{\pm 0.06}}$ | $1.07_{\pm 0.05}$ |
| $\text{PROSE}_{Full+ICL}$ | $\underline{1.34_{\pm 0.09}}$ | $\underline{1.34_{\pm 0.07}}$ | $1.39_{\pm 0.09}$ | $\mathbf{1.65_{\pm 0.05}}$ | $1.37_{\pm 0.06}$ | $\mathbf{1.49_{\pm 0.04}}$ |

Table 2: PROSE's performance on the two tasks measured by the quality of inferred preferences (Pref. Sim.) and preference compliance (PPCM) compared against no preference conditioning (NPC), true preference generation (Oracle), in-context learning (ICL), CIPHER (Gao et al., 2024), and ablations over PROSE's components. Results are the mean and pooled standard error across the four LLMs and five seeds. Best results are bolded, second best are underlined.

well an agent's writing aligns with the user's true preferences (see Appendix B.2). Both performance measures use LLM-as-a-Judge to assess the similarity between the true and inferred preferences, and between the true preferences and the agent's generations.

The agent aligns itself with four (email) or five (summarization) users with five demonstrations per user. Performance is evaluated per task as the mean across all demonstrations, users, and task type. Each task is run over five seeds (standard error is reported over the seeds). The ground truth user preferences by task and task type are in Appendix D (Table 9). The performance of four LLMs is reported and compared: `Qwen2.5-7B-Instruct`, `Qwen2.5-72B-Instruct`, `GPT-4o-mini`, and `GPT-4o`(Yang et al., 2024; OpenAI, 2023). For all LLMs, $S$ and $v$ are determined via a hyperparameter sweep over $v \in 0, 0.25, 0.5, 0.75, 1$ and $S \in 2, 3, 4, 5$. In our experiments $S = 5$ for all LLMs, and $v = 0.25$ for `Qwen2.5-7B-Instruct`, $v = 0.5$ for both `GPT-4o`models, and $v = 0.75$ for `Qwen2.5-72B-Instruct`. For all experiments GPT-4o is used as a synthetic human. The synthetic human prompts can be found in Appendix F.2.

## 5.1. Research Questions

**RQ1: Does iterative refinement improve performance?**

We consider three variants of PROSE: (1) $\text{PROSE}_u$ infers $\hat{p}_{\text{desc}}$ given only the user's demonstration $w_u$; (2) $\text{PROSE}_{u,a}$ infers $\hat{p}_{\text{desc}}$ given the user's demonstration $w_u$ and the initial assistant generation $w_a^{s=0}$; and (3) $\text{PROSE}_{u,a,S>1}$ refines $\hat{p}_{\text{desc}}$ over $\leq S$ inference steps given the user's demonstration $w_u$ and the initial assistant generation $w_a^{s=0}$. $\text{PROSE}_{Full}$ refines $\hat{p}_{\text{desc}}$ over $\leq S$ inference steps given the user's demonstration $w_u$ and the iteratively refined assistant generations $w_a^{s\in[0,S]}$. Comparing $\text{PROSE}_u$ and $\text{PROSE}_{u,a}$ measures the effect of comparing assistant generations to the user demonstration when inferring preferences. The differences between the $\text{PROSE}_{u,a}$ and $\text{PROSE}_{u,a,S>1}$ quantifies the role of increasing the number of refinement steps. Lastly, comparing $\text{PROSE}_{u,a,S>1}$ and the the complete PROSE algorithm $\text{PROSE}_{Full}$ clarifies the effects of comparing the user demonstration to the assistant's generation conditioned on the latest inferred preference description.

**RQ2: Does filtering preferences that are not relevant to multiple user demonstrations improve performance?**

To answer this question, we evaluate a variant, $\text{PROSE}_{NV}$, that does not use the preference consistency verification step Section 3.2.

**RQ3: Is conditioning on preferences better than conditioning on demonstrations?**

To answer this question, we compare PROSE, CIPHER (Gao et al., 2024), and ICL on all tasks, task types, and user profiles. We additionally combine the PROSE and ICL to

measure the extent to which they are complementary.

## 5.2. Baselines

In addition to the PROSE baselines outlined Section 5.1, we implement the following models.

We implement CIPHER-1 and CIPHER-5 (Gao et al., 2024), and an in-context learning (ICL) agent using previously observed user demonstrations. The CIPHER baselines are adapted to learn from PLUME's user demonstrations instead of user edits.

We then implement three additional baselines. An agent that solves the task with no preference conditioning (NPC), providing a lower-bound of performance. An oracle agent (Oracle) that receives access to the user's true preference, providing an upper bound of performance, and a variation of PROSE that is conceptually equivalent to CIPHER, $PROSE_{CE}$, but uses PROSE's improved prompt templates. $PROSE_{CE}$ uses a single LLM generation, a single inference step, and uses no preference consistency verification.

## 6. Results and Discussion

We present our main PLUME results in Table 2. Results on PRELUDE can be found in Appendix C.3. To compare tasks on generation quality with metrics on different scales, we use a percentile score, where 0% corresponds to the no preference conditioning baseline (NPC) and 100% to the Oracle baseline. Percent improvements are reported as the difference in scores on this scale. Overall, $PROSE_{Full}$ outperforms $PROSE_{CE}$ by 12%, and CIPHER by 33%.

**RQ1.** In our first question, we set out to verify whether generating iterative candidate trajectories is beneficial to inferring preferences. Comparing PROSE to its ablated versions on the action/generation quality metric (PPCM), shows that each component of the iterative refinement process improves performance. Comparing PROSE with no comparison generation — $PROSE_u$ — to PROSE with a single LLM-generated comparison generation — $PROSE_{u,a}$ — we observe that providing the comparison generations is beneficial when inferring preferences (3.8% mean improvement). This result supports the algorithmic decisions in (Gao et al., 2024; Peng et al., 2024). Allowing for multiple refinement steps provides a further increase in performance (Table 2: $PROSE_{u,a}$ vs. $PROSE_{u,a,S>1}$, 7.8% mean improvement). This can be explained by the LLM having more chances to infer correct preferences. Lastly, when comparing $PROSE_{u,a,S>1}$ to $PROSE_{Full}$ we see another 3.2% improvement. This highlights the benefits of updating candidates after each inference step using the newly inferred preferences. In all, iterative refinement provides a mean improvement of 14.8%.

**RQ2.** We investigate the benefit of verifying preferences by comparing PROSE to $PROSE_{NV}$. Here, we see a modest but consistent of 1.5% and 1.7% for Pref. Sim. and PPCM respectively when using preference consistency verification.

**RQ3.** While on average across LLMs PROSE outperforms CIPHER and all PROSE ablations, ICL outperforms PROSE. However, Figure 2 shows that PROSE's performance scales better with the quality of the underlying LLM (e.g., Qwen2.5-72B-Instruct vs. GPT-4o) than all baselines except Oracle. Notably, when using GPT-4o, PROSE outperforms ICL (1.35 vs 1.32 task mean Appendix C.2). We further investigate the benefits and limitations of PROSE and the learning baselines by comparing the performance across preference sets (Figure 3), and find that ICL excels on sets with the strongest structural preferences (e.g., Chat Forum Posts which includes "write in the style of a tweet"). In contrast, PROSE excels on the preference sets requiring a more nuanced understanding of tone (e.g., Paper Review, which includes "be sharply critical"). From examining logs, we notice that the LLMs are less adept at inferring encompassing structural preferences and often try to capture these preferences using multiple relevant, but imperfect preferences (e.g., "use emojis for emphasis", "use 1-2 specific hashtags") As PROSE and ICL seem to have complementary strengths, we combine the two ($PROSE_{Full+ICL}$) for a gain of 7.8%, 8.9%, and 51.1% over PROSE, ICL, and CIPHER when using GPT-4o as the agent's LLM.

**Human Evaluation** To further validate the effectiveness of PROSE, we ran human evaluation with 16 participants (3 are ML researchers; 9 women and 7 men; age in [19, 58]). Participants completed a within subjects AB test comparing PLUME+ICL generations to ICL generations and PLUME+ICL generations to CIPHER generations. Participants evaluated the final LLM generations (i.e. the generation after seeing all previous demonstrations) across all five seeds for two different preference sets for the email task and two different preference sets for the summarization task. This leads to a total of 20 survey items per method comparison. We used the responses to compute a win rate for PLUME+ICL compared to each of ICL and CIPHER. For PLUME+ICL versus ICL, we see an average win rate of 69.4%. For PLUME+ICL versus CIPHER, we see an average win rate of 91.8%. The human evaluation results are in line with our synthetic evaluation results and support the effectiveness of the synthetic evaluation.

**Discussion.** Our results demonstrate that using iterative refinement and consistency verification improves over CIPHER in terms of preference description quality, generation quality, and performance stability (i.e. performance increases with the number of demonstrations, see Figure 6). Additionally the performance difference between CIPHER

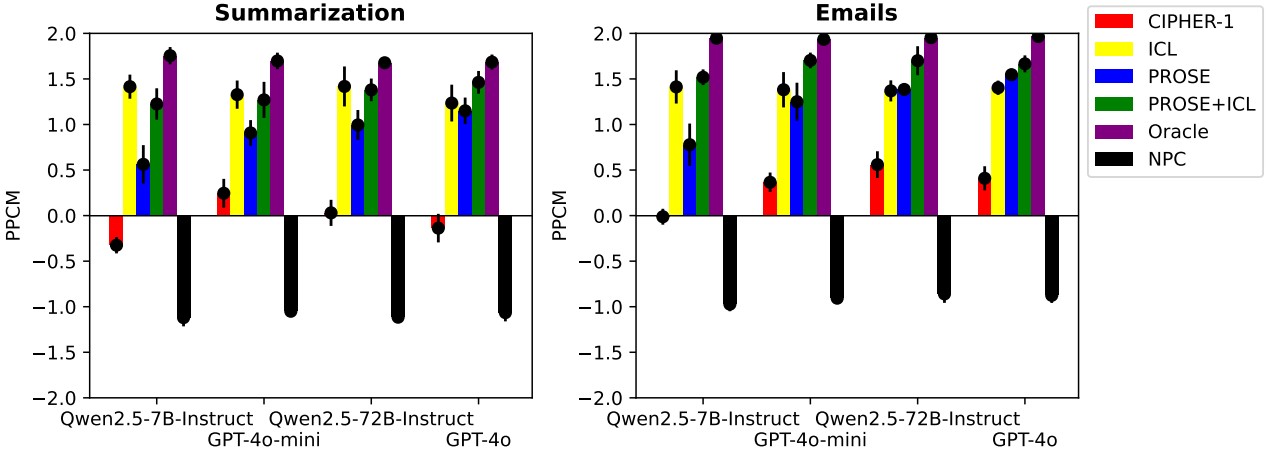

Figure 2: Preference compliance performance (PPCM) for CIPHER-1, in-context learning (ICL), PROSE, Oracle, and no preferences (NPC) for different preference-inferring LLMs. The LLMs are sorted by MMLU performance: `Qwen2.5-7B-Instruct` = 74.2, `GPT-4o-mini` = 82, `Qwen2.5-72B-Instruct` = 86.1, and `GPT-4o` = 88.7. `GPT-4o` is the proxy human with mean and standard error reported over 5 seeds.

and PROSE$_{CE}$ highlights the impact of our prompt-tuning efforts. In this regard, PLUME's prompts (Appendix F) can serve as a valuable starting point for extensions to other tasks, however, task-specific adaptations should be made.

Our results suggest that consistency verification provides only a modest improvement to PROSE. Therefore, to better understand its impact, we examine the learning logs and find that consistency verification effectively prunes irrelevant preferences—e.g.,"be concise and direct"— and preferences that overfit to specific passages— e.g., " include personal details about characters". However, the pruned preferences typically have minimal impact on the performance metrics as they rarely contradict the true preferences. Moreover, the pruned preferences do not drastically alter the generations as the orthogonal preferences often match the LLM's default behavior while the overfit preferences become irrelevant and ignored. As such, the current metrics have difficulty measuring the presence of these irrelevant preferences. Nevertheless, we believe it is valuable to prune the irrelevant preferences, as they reduce the number of tokens required.

We find PROSE is competitive with and complementary to ICL while providing several advantages: (1) preferences are easier to interact with than a dataset of in-context examples as a user can view and modify the inferred preferences, (2) at inference time, PROSE requires approximately $\frac{1}{10}$ of the prompt tokens, and (3) the inferred preference description can benefit a wider range of tasks (e.g. human-agent collaboration (Liu et al., 2024), sample efficient imitation/reinforcement learning, and generating personalized preference pairs for RLAIF (Sun et al., 2024)).

Lastly, while developing PROSE, we learned the importance of phrasing the preference description in the LLM's "own words". We initially sorted the preference components by length before aggregation, however, this led to an average performance drop of 11% across tasks relative to keeping the LLM's order for the preference components. This finding is inline with other work that shows that LLMs are sensitive to the order of list items (Pezeshkpour & Hruschka, 2024; Aroca-Ouellette et al., 2021). We believe future work investigating the impact of the ordering may yield useful insights.

### 6.1. Limitations and Future Work

While PROSE and PLUME provide a number improvements, their limitations and challenges provide interesting avenues for future work. First, in this paper we focus on learning with the fewest user demonstrations possible. However another aspect of efficiency is the total number of tokens, and adding more refinement and preference consistency verification steps increases the number of tokens used. In our experiments, PROSE$_{Full}$ used 5.87x (prompt) and 6.07x (generated) more tokens on average than PROSE$_{CE}$. Given the monetary and environmental cost of LLMs, reducing the number of tokens while retaining performance is an important area for improvement. Lastly, a full-scale human trial would provide a greater understanding of the benefits and limitations of the proposed method. We look forward to investigating this more closely in future work.

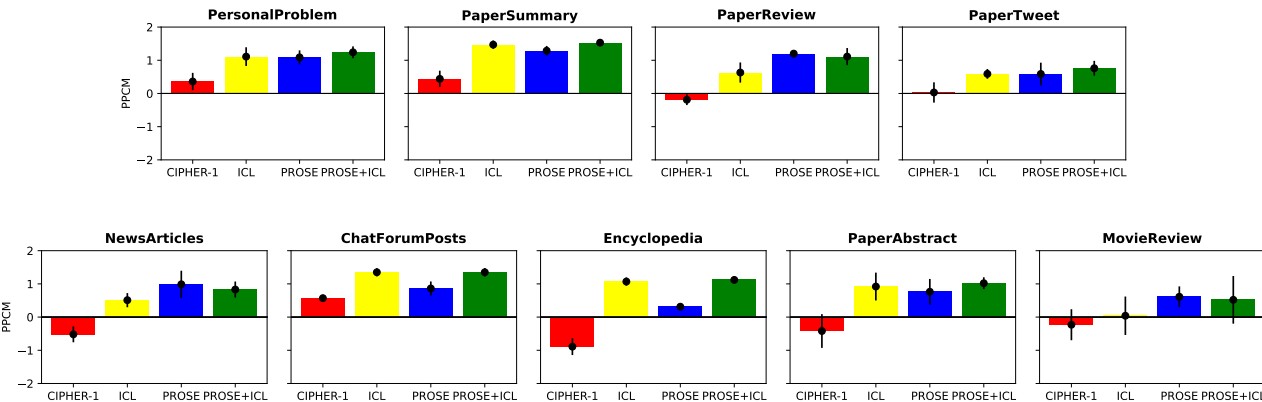

Figure 3: Generation quality (PPCM) for PROSE, CIPHER-1, in-context learning (ICL), and PROSE+ICL by **Email** (top) and **Summary** (bottom). `GPT-4o` is the agent's LLM with mean and standard error reported over 5 seeds.

## 7. Conclusion

In this paper, we present a novel algorithm, PROSE, and a new benchmark, PLUME. PROSE's two novel contributions guide an LLM to better infer preferences from user demonstrations by: (1) iteratively refining preferences by conditioning an LLM on each refinement step's updated preference description to see its impact, and (2) decompose preferences into components and verify the components against relevant user demonstrations. We demonstrate that the proposed method improves an LLM's ability to align to a user by as much as 33%.

## Impact Statement

The proposed method allows for greater personalization of assistive agents. However inferring a user's preferences could be seen as an invasion of privacy. These methods should be applied only with explicit consent from human users.

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

# A. Algorithm

---

**Algorithm 1** Assistant Task Completion

---

**Require:** $x_{\text{task}}$ {Task instance}
 1: Initialize empty preference set: $\hat{P}_c \leftarrow \varnothing$
 2: Retrieve relevant examples:
 3:     $E \leftarrow$ *get_relevant_examples*(`interaction memory`)
 4: **for** each $e \in E$ **do**
 5:     $\hat{P}_c \leftarrow \hat{P}_c \cup e.\hat{p_c}$
 6: **end for**
 7: Aggregate condense preferences:
 8:     $\hat{p}_{desc} \leftarrow$ `generate`($\text{llm}, x_{\text{aggregate}}, \hat{P}_c$)
 9: Sample agent generation:
10:     $w_a^0 \leftarrow$ `generate`($\text{llm}, x_{\text{task}}, \hat{p}_{desc}$)
11: **Return** Completed generation $w_a^0$, preference description $\hat{p}_{desc}$, and relevant examples $E$

---

---

**Algorithm 2** PROSE: Preference Reasoning by Observing and Synthesizing Examples

---

**Require:** $x_{\text{task}}$ {Task instance}
**Require:** $w_u$ {User demonstration}
**Require:** $w_a^0$ {Agent generation}
**Require:** $\hat{p}_{desc}$ {Preference description}
**Require:** $E$ {relevant Examples}
 1: Initialize $\hat{P}_c \leftarrow \texttt{generate}(\texttt{llm}, x_{\text{breakdown}}, \hat{p}_{\text{desc}}^0)$
 2: **for** each $s \in [0, S]$ **do**
 3:     **if** $w_a^s = w_u$ **then**
 4:         Stop refinement
 5:     **else**
 6:         Refine preferences:
 7:             $\hat{p}_{\text{desc}}^{s+1} = \texttt{generate}(\texttt{llm}, x_{\text{update}}, \hat{p}_{\text{desc}}^s, w_u, w_a^s)$
 8:         Decompose preference:
 9:             $\hat{P}_c \leftarrow \hat{P}_c \cup \texttt{generate}(\texttt{llm}, x_{\text{breakdown}}, \hat{p}_{\text{desc}}^s)$
10:         Generate new candidate generation:
11:             $w_a^s \leftarrow \texttt{generate}(\texttt{llm}, x_{\text{task}}, \hat{p}_{desc})$
12:     **end if**
13: **end for**
14: Initialize empty verification score list:
15:     $v_{scores} \leftarrow \varnothing$
16: **for** each $\hat{p}_c$ in $\hat{P}_c$ **do**
17:     **for** each $e \in E$ **do**
18:         Verify preference against demonstration:
19:             $v_{scores} \leftarrow v_{scores} \cup \texttt{generate}(\texttt{llm}, x_{\text{verification}}, e.w_u^i, \hat{p}_{\text{desc}}^s)$
20:     **end for**
21:     **if** $\texttt{mean}(v_{scores}) < v$ **then**
22:         Discard $\hat{p}_c$ from $\hat{P}_c$
23:     **end if**
24: **end for**
25: Add $(x_{\text{task}}, w_u^i, \hat{P}_c)$ to $\texttt{interaction memory}$

---

# B. Metric Definitions

## B.1. Preference Inference Quality

**Preference Description Length** As conditioning on unnecessary tokens when generating responses aligned with user preferences is undesirable, we measure the number of tokens in the preference description. The preference length (Pref Len) is the number of characters used to describe a user's preferences, which is highly correlated with the number of tokens required.

**Preference Similarity** To assess the similarity between the inferred preferences and the ground truth preferences, the human proxy (GPT-4o in this paper) is prompted to evaluate how similar each inferred preference is to each ground truth preference following:

$$\text{Preference Similarity} = \text{llm\_judge}(\text{true}, \text{inferred}), \tag{1}$$

where true is the true preferences (see Appendix D Table 9), inferred is the inferred preference description, and llm_judge is a function that prompts the human proxy LLM to evaluate how well a given inferred preference aligns with the true preference on a scale of 0 to +4 (see Appendix Figure 4).

---

**System Prompt**

You are an experienced editor that is evaluating how similar writing preferences are.

**User Prompt**

You received the following description of a user's writing preferences:
""""""

Inferred preference: `<inferred_preference_i>`
""""""

"How similar are the inferred preferences to the true writing preferences below?
True preference: `<true_preference_i>`?
Analyze how the preferences would impact a user's writing. After reasoning, select one of the following options:
extremely similar, very similar, moderately similar, slightly similar, not at all similar
Your final selection should be on a new line prefaced with "Verdict:"

---

Figure 4: **LLM-as-a-Judge prompts** to assess the similarity between the true and inferred preferences. The system prompt is prepended to the user prompt following the LLM's chat template. "`<...>`" indicates that the text is formatted from a variable. `inferred_preference_i` one of the inferred preferences. `true_preference_i` refers to one of the $k$ true preferences that the user has.

## B.2. Generation Quality

To assess the quality of the preference conditioned LLM's generations, the human proxy (GPT-4o in this paper) is prompted to evaluate how well the given generation complies with the each of the ground truth user preferences. The generation quality is then compute as the mean score over ground truth user preference components following:

$$\text{PPCM} = \frac{\sum_i^{|\text{true}|} \text{llm\_judge}(\text{true}_i, \text{assistant\_attempt})}{|\text{true}|}, \tag{2}$$

where true is the set of true preferences (see Appendix D Table 9), assistant_attempt is the assistant's summary or email, and llm_judge is a function that prompts the human proxy LLM to evaluate how well a given assistant solution aligns with the true preference on a scale of -2 to +2 (see Figure 5).

The prompt used by the LLM-as-a-Judge for generation quality evaluation are shown in Appendix Figure 5.

**System Prompt**

You are an experienced editor that is evaluating writing samples.

**User Prompt**

You received the following {`summary` | `email`}:
"""

`<agent_completion>`
"""

Does the above {`summary` | `email`} exhibit the following preference: `<true_preference_i>`?
Identify, analyze, and reason about specific excerpts that show similarities or contradictions of underlying preferences.
After reasoning, select one of the following options:
clearly exhibits, somewhat exhibits, neither exhibits nor contradicts, somewhat contradicts, clearly contradicts
Your final selection should be on a new line prefaced with "Verdict:"

Figure 5: **LLM-as-a-Judge prompts** for the per preference-component match metric (PPCM) used in the **PLUME environment**. The system prompt is prepended to the user prompt following the LLM's chat template. "`<...>`" indicates that the text is formatted from a variable. `agent_completion` refers to the agent's article summary or email, depending on the sub-task. `true_preference_i` refers to one of the $k$ true preferences that the user has.

# C. Extended Results

Additional results tables and figures discussed in the main body of the paper.

### C.1. Metric Correlation

The metric correlation results for the assistive writing tasks (Table 3).

| Metric | PRELUDE | | | | PRELUDE $_{NoEdit}$ | | | | PLUME | | | |
|---|---|---|---|---|---|---|---|---|---|---|---|---|
| | Acc. | B.Score | P. Len. | P. Sim. | Acc. | B.Score | P. Len. | P. Sim. | Acc. | B.Score | P. Len. | P. Sim. |
| Summarization | | | | | | | | | | | | |
| L-dist | -0.47 | -0.53 | -0.43 | -0.46 | -0.08 | -0.14 | 0.01 | -0.18 | 0.02 | -0.13 | -0.21 | -0.10 |
| ln-L-dist | -0.50 | -0.57 | -0.48 | -0.51 | -0.22 | -0.36 | -0.23 | -0.42 | -0.08 | -0.33 | -0.40 | -0.36 |
| PPCM | 0.49 | **0.63** | 0.53 | 0.54 | 0.51 | **0.65** | 0.54 | 0.58 | 0.34 | 0.65 | **0.71** | **0.71** |
| Emails | | | | | | | | | | | | |
| L-dist | -0.26 | -0.32 | -0.27 | -0.26 | -0.10 | -0.31 | -0.36 | -0.27 | -0.11 | -0.19 | -0.25 | -0.19 |
| ln-L-dist | -0.23 | -0.31 | -0.27 | -0.26 | -0.08 | -0.34 | -0.42 | -0.28 | -0.12 | -0.34 | -0.41 | -0.38 |
| PPCM | 0.18 | 0.30 | **0.43** | 0.18 | 0.20 | 0.32 | **0.45** | 0.19 | 0.48 | **0.79** | 0.74 | **0.79** |
| Across Both Tasks | | | | | | | | | | | | |
| L-dist | -0.43 | -0.43 | -0.31 | -0.39 | -0.09 | -0.17 | -0.03 | -0.20 | 0.01 | -0.15 | -0.21 | -0.11 |
| ln-L-dist | -0.45 | -0.45 | -0.32 | -0.42 | -0.18 | -0.27 | -0.08 | -0.32 | -0.09 | -0.32 | -0.39 | -0.35 |
| PPCM | 0.42 | **0.48** | 0.37 | 0.42 | 0.45 | 0.52 | 0.39 | **0.46** | 0.39 | 0.68 | 0.71 | **0.73** |

Table 3: Pearson R correlation between preference similarity metrics and generated writing similarity metrics broken down by task (summarization vs email). For Levenshtein distance (L-dist) and length-normalized Levenshtein distance (ln-L-dist) lower is better, so inverse correlation is expected. For PPCM, Acc. (Accuracy), B.SCore (BertScore), P. Len (Preference Description Length), and P. Sim. (Preference Description Similarity) higher is better. Best correlation in each framework is bold. Best overall correlation is underlined. See Appendix B for a full description of each metric.

## C.2. Baselines and Ablations per LLM

The baseline and PROSE ablation results for `Qwen2.5-7b-Instruct`, `Qwen2.5-72b-Instruct`, `GPT-4o-mini`, and `GPT-4o`. The results in Table 2 are averaged across these four LLMs and results reported in this section.

| Method | Summarization | | | Emails | | |
|---|---|---|---|---|---|---|
| | Pref Len | Pref. Sim. | PPCM | Pref Len | Pref. Sim. | PPCM |
| No Learning Baselines | | | | | | |
| NPC | $0.00_{\pm0.00}$ | $0.00_{\pm0.00}$ | $-1.12_{\pm0.04}$ | $0.00_{\pm0.00}$ | $0.00_{\pm0.00}$ | $-0.97_{\pm0.04}$ |
| Oracle | $120.80_{\pm0.00}$ | $3.88_{\pm0.05}$ | $1.75_{\pm0.04}$ | $118.50_{\pm0.00}$ | $3.90_{\pm0.06}$ | $1.95_{\pm0.01}$ |
| Learning Baselines | | | | | | |
| ICL | $6237.54_{\pm1163.61}$ | $0.00_{\pm0.00}$ | $\mathbf{1.42_{\pm0.06}}$ | $6754.73_{\pm975.36}$ | $0.00_{\pm0.00}$ | $1.41_{\pm0.08}$ |
| CIPHER-1 | $50.40_{\pm1.68}$ | $0.90_{\pm0.04}$ | $-0.33_{\pm0.04}$ | $51.55_{\pm0.93}$ | $1.32_{\pm0.10}$ | $-0.01_{\pm0.04}$ |
| CIPHER-5 | $\mathbf{49.78_{\pm1.86}}$ | $\mathbf{1.04_{\pm0.05}}$ | $-0.11_{\pm0.07}$ | $\mathbf{49.04_{\pm2.46}}$ | $\mathbf{1.66_{\pm0.05}}$ | $-0.01_{\pm0.06}$ |
| PROSE Ablations | | | | | | |
| PROSE$_{CE}$ | $306.05_{\pm15.72}$ | $0.59_{\pm0.03}$ | $0.07_{\pm0.06}$ | $390.29_{\pm20.64}$ | $0.99_{\pm0.05}$ | $0.62_{\pm0.06}$ |
| PROSE$_u$ | $274.05_{\pm20.33}$ | $0.45_{\pm0.08}$ | $-0.02_{\pm0.05}$ | $353.77_{\pm33.87}$ | $0.84_{\pm0.05}$ | $0.16_{\pm0.15}$ |
| PROSE$_{u,a}$ | $291.45_{\pm15.41}$ | $0.61_{\pm0.11}$ | $0.03_{\pm0.15}$ | $301.27_{\pm26.19}$ | $1.00_{\pm0.09}$ | $0.45_{\pm0.04}$ |
| PROSE$_{u,a,S>1}$ | $574.73_{\pm27.75}$ | $0.77_{\pm0.06}$ | $0.40_{\pm0.11}$ | $611.92_{\pm32.87}$ | $1.14_{\pm0.06}$ | $0.83_{\pm0.05}$ |
| PROSE$_{NV}$ | $745.49_{\pm30.03}$ | $0.90_{\pm0.07}$ | $0.48_{\pm0.10}$ | $753.11_{\pm75.90}$ | $0.97_{\pm0.11}$ | $0.72_{\pm0.04}$ |
| PROSE$_{Full}$ | $604.69_{\pm31.19}$ | $0.89_{\pm0.10}$ | $0.56_{\pm0.09}$ | $698.34_{\pm44.28}$ | $0.91_{\pm0.09}$ | $0.78_{\pm0.10}$ |
| PROSE$_{Full+ICL}$ | $6829.38_{\pm1159.22}$ | $0.76_{\pm0.07}$ | $1.23_{\pm0.08}$ | $7435.39_{\pm980.26}$ | $0.96_{\pm0.06}$ | $\mathbf{1.52_{\pm0.04}}$ |

Table 4: `Qwen2.5-7b-Instruct` + PROSE's performance on the two tasks measured by the quality of inferred preferences (Pref. Sim.) and preference compliance (PPCM) compared against no preference generation (NPC), true preference generation (Oracle), in-context learning (ICL), CIPHER (Gao et al., 2024), and ablations over PROSE's components. Results are the mean and standard error across five seeds. Best non-Oracle results per task are bolded.

| Method | Summarization | | | Emails | | |
|---|---|---|---|---|---|---|
| | Pref Len | Pref. Sim. | PPCM | Pref Len | Pref. Sim. | PPCM |
| No Learning Baselines | | | | | | |
| NPC | $0.00_{\pm 0.00}$ | $0.00_{\pm 0.00}$ | $-1.11_{\pm 0.02}$ | $0.00_{\pm 0.00}$ | $0.00_{\pm 0.00}$ | $-0.86_{\pm 0.04}$ |
| Oracle | $120.80_{\pm 0.00}$ | $3.88_{\pm 0.08}$ | $1.68_{\pm 0.03}$ | $118.50_{\pm 0.00}$ | $3.85_{\pm 0.06}$ | $1.95_{\pm 0.01}$ |
| Learning Baselines | | | | | | |
| ICL | $6237.54_{\pm 1163.61}$ | $0.00_{\pm 0.00}$ | $\mathbf{1.42_{\pm 0.10}}$ | $6754.73_{\pm 975.36}$ | $0.00_{\pm 0.00}$ | $1.37_{\pm 0.05}$ |
| CIPHER-1 | $\mathbf{88.36_{\pm 1.91}}$ | $1.26_{\pm 0.04}$ | $0.03_{\pm 0.06}$ | $\mathbf{83.01_{\pm 2.03}}$ | $\mathbf{1.82_{\pm 0.07}}$ | $0.56_{\pm 0.07}$ |
| CIPHER-5 | $141.23_{\pm 6.64}$ | $1.22_{\pm 0.05}$ | $-0.06_{\pm 0.06}$ | $141.45_{\pm 8.35}$ | $1.60_{\pm 0.08}$ | $0.34_{\pm 0.05}$ |
| PROSE Ablations | | | | | | |
| $PROSE_{CE}$ | $359.01_{\pm 23.18}$ | $1.25_{\pm 0.09}$ | $0.62_{\pm 0.10}$ | $375.76_{\pm 6.64}$ | $1.56_{\pm 0.10}$ | $1.06_{\pm 0.05}$ |
| $PROSE_u$ | $333.35_{\pm 13.53}$ | $1.35_{\pm 0.13}$ | $0.65_{\pm 0.09}$ | $369.00_{\pm 21.15}$ | $1.29_{\pm 0.07}$ | $0.87_{\pm 0.09}$ |
| $PROSE_{u,a}$ | $304.55_{\pm 16.14}$ | $1.41_{\pm 0.07}$ | $0.52_{\pm 0.09}$ | $352.71_{\pm 18.17}$ | $1.75_{\pm 0.09}$ | $1.17_{\pm 0.04}$ |
| $PROSE_{u,a,S>1}$ | $635.88_{\pm 39.19}$ | $1.42_{\pm 0.04}$ | $0.84_{\pm 0.05}$ | $829.24_{\pm 37.09}$ | $1.56_{\pm 0.10}$ | $1.39_{\pm 0.08}$ |
| $PROSE_{NV}$ | $937.37_{\pm 74.59}$ | $1.57_{\pm 0.05}$ | $0.97_{\pm 0.11}$ | $1004.21_{\pm 30.02}$ | $1.36_{\pm 0.13}$ | $1.32_{\pm 0.11}$ |
| $PROSE_{Full}$ | $628.28_{\pm 36.94}$ | $\mathbf{1.60_{\pm 0.11}}$ | $0.99_{\pm 0.07}$ | $880.98_{\pm 22.95}$ | $1.55_{\pm 0.06}$ | $1.38_{\pm 0.03}$ |
| $PROSE_{Full+ICL}$ | $6912.28_{\pm 1171.86}$ | $1.41_{\pm 0.10}$ | $1.38_{\pm 0.05}$ | $7624.14_{\pm 973.68}$ | $1.45_{\pm 0.13}$ | $\mathbf{1.70_{\pm 0.07}}$ |

Table 5: `Qwen2.5-72b-Instruct` + PROSE's performance on the two tasks measured by the quality of inferred preferences (Pref. Sim.) and preference compliance (PPCM) compared against no preference conditioning (NPC), true preference conditioning (Oracle), in-context learning (ICL), CIPHER (Gao et al., 2024), and ablations over PROSE's components. Results are the mean and standard error across five seeds. Best non-Oracle results per task are bolded.

| Method | Summarization | | | Emails | | |
|---|---|---|---|---|---|---|
| | Pref Len | Pref. Sim. | PPCM | Pref Len | Pref. Sim. | PPCM |
| No Learning Baselines | | | | | | |
| NPC | $0.00_{\pm 0.00}$ | $0.00_{\pm 0.00}$ | $-1.05_{\pm 0.02}$ | $0.00_{\pm 0.00}$ | $0.00_{\pm 0.00}$ | $-0.91_{\pm 0.02}$ |
| Oracle | $120.80_{\pm 0.00}$ | $3.84_{\pm 0.08}$ | $1.70_{\pm 0.04}$ | $118.50_{\pm 0.00}$ | $3.95_{\pm 0.05}$ | $1.93_{\pm 0.01}$ |
| Learning Baselines | | | | | | |
| ICL | $6237.54_{\pm 1163.61}$ | $0.00_{\pm 0.00}$ | $\mathbf{1.33_{\pm 0.07}}$ | $6754.73_{\pm 975.36}$ | $0.00_{\pm 0.00}$ | $1.38_{\pm 0.08}$ |
| CIPHER-1 | $138.91_{\pm 3.67}$ | $1.44_{\pm 0.03}$ | $0.24_{\pm 0.07}$ | $148.69_{\pm 1.51}$ | $1.79_{\pm 0.03}$ | $0.37_{\pm 0.05}$ |
| CIPHER-5 | $\mathbf{74.93_{\pm 2.42}}$ | $1.26_{\pm 0.10}$ | $-0.05_{\pm 0.08}$ | $\mathbf{78.09_{\pm 2.17}}$ | $\mathbf{1.74_{\pm 0.08}}$ | $0.30_{\pm 0.10}$ |
| PROSE Ablations | | | | | | |
| $PROSE_{CE}$ | $384.12_{\pm 12.65}$ | $1.44_{\pm 0.07}$ | $0.67_{\pm 0.05}$ | $412.09_{\pm 12.57}$ | $1.57_{\pm 0.03}$ | $1.03_{\pm 0.08}$ |
| $PROSE_u$ | $254.81_{\pm 15.59}$ | $1.55_{\pm 0.12}$ | $0.47_{\pm 0.11}$ | $364.86_{\pm 13.18}$ | $1.40_{\pm 0.10}$ | $1.01_{\pm 0.10}$ |
| $PROSE_{u,a}$ | $321.09_{\pm 8.23}$ | $\mathbf{1.76_{\pm 0.11}}$ | $0.70_{\pm 0.07}$ | $375.89_{\pm 24.73}$ | $1.75_{\pm 0.08}$ | $1.24_{\pm 0.05}$ |
| $PROSE_{u,a,S>1}$ | $551.56_{\pm 16.27}$ | $1.46_{\pm 0.11}$ | $0.82_{\pm 0.13}$ | $745.58_{\pm 29.95}$ | $1.50_{\pm 0.08}$ | $1.30_{\pm 0.13}$ |
| $PROSE_{NV}$ | $699.89_{\pm 22.87}$ | $1.48_{\pm 0.07}$ | $0.94_{\pm 0.05}$ | $798.89_{\pm 37.11}$ | $1.30_{\pm 0.10}$ | $1.16_{\pm 0.07}$ |
| $PROSE_{Full}$ | $575.88_{\pm 20.20}$ | $1.58_{\pm 0.08}$ | $0.91_{\pm 0.06}$ | $699.25_{\pm 33.42}$ | $1.60_{\pm 0.07}$ | $1.25_{\pm 0.09}$ |
| $PROSE_{Full+ICL}$ | $6795.69_{\pm 1167.68}$ | $1.42_{\pm 0.08}$ | $1.27_{\pm 0.09}$ | $7542.95_{\pm 977.22}$ | $1.52_{\pm 0.08}$ | $\mathbf{1.70_{\pm 0.04}}$ |

Table 6: `GPT-4o-mini` PROSE's performance on the two tasks measured by the quality of inferred preferences (Pref. Sim.) and preference compliance (PPCM) compared against no preference conditioned (NPC), true preference generation (Oracle), in-context learning (ICL), CIPHER (Gao et al., 2024), and ablations over PROSE's components. Results are the mean and standard error across five seeds. Best non-Oracle results per task are bolded.

| Method | Summarization | | | Emails | | |
|---|---|---|---|---|---|---|
| | Pref Len | Pref. Sim. | PPCM | Pref Len | Pref. Sim. | PPCM |
| No Learning Baselines | | | | | | |
| NPC | $0.00_{\pm 0.00}$ | $0.00_{\pm 0.00}$ | $-1.06_{\pm 0.04}$ | $0.00_{\pm 0.00}$ | $0.00_{\pm 0.00}$ | $-0.88_{\pm 0.04}$ |
| Oracle | $120.80_{\pm 0.00}$ | $3.84_{\pm 0.08}$ | $1.69_{\pm 0.04}$ | $118.50_{\pm 0.00}$ | $3.85_{\pm 0.06}$ | $1.97_{\pm 0.01}$ |
| Learning Baselines | | | | | | |
| ICL | $6237.54_{\pm 1163.61}$ | $0.00_{\pm 0.00}$ | $1.24_{\pm 0.09}$ | $6754.73_{\pm 975.36}$ | $0.00_{\pm 0.00}$ | $1.40_{\pm 0.04}$ |
| CIPHER-1 | $60.80_{\pm 1.24}$ | $1.23_{\pm 0.04}$ | $-0.14_{\pm 0.07}$ | $67.62_{\pm 1.37}$ | $1.75_{\pm 0.07}$ | $0.41_{\pm 0.06}$ |
| CIPHER-5 | $\mathbf{56.03_{\pm 1.52}}$ | $1.43_{\pm 0.07}$ | $-0.11_{\pm 0.13}$ | $\mathbf{56.81_{\pm 2.00}}$ | $1.76_{\pm 0.05}$ | $0.38_{\pm 0.06}$ |
| PROSE Ablations | | | | | | |
| $\text{PROSE}_{\text{CE}}$ | $314.11_{\pm 19.75}$ | $1.63_{\pm 0.05}$ | $0.66_{\pm 0.11}$ | $342.27_{\pm 12.98}$ | $1.71_{\pm 0.08}$ | $1.18_{\pm 0.10}$ |
| $\text{PROSE}_{u}$ | $228.11_{\pm 11.47}$ | $1.84_{\pm 0.10}$ | $0.77_{\pm 0.14}$ | $303.74_{\pm 7.32}$ | $1.84_{\pm 0.14}$ | $1.30_{\pm 0.06}$ |
| $\text{PROSE}_{u,a}$ | $249.26_{\pm 11.70}$ | $1.62_{\pm 0.12}$ | $0.70_{\pm 0.10}$ | $317.54_{\pm 6.48}$ | $1.82_{\pm 0.09}$ | $1.29_{\pm 0.10}$ |
| $\text{PROSE}_{u,a,S>1}$ | $428.30_{\pm 18.53}$ | $1.83_{\pm 0.17}$ | $0.92_{\pm 0.08}$ | $534.29_{\pm 18.92}$ | $1.79_{\pm 0.06}$ | $1.33_{\pm 0.01}$ |
| $\text{PROSE}_{\text{NV}}$ | $489.66_{\pm 13.24}$ | $1.94_{\pm 0.06}$ | $1.07_{\pm 0.12}$ | $513.60_{\pm 11.57}$ | $\mathbf{1.90_{\pm 0.05}}$ | $1.50_{\pm 0.07}$ |
| $\text{PROSE}_{\text{Full}}$ | $446.73_{\pm 10.71}$ | $\mathbf{1.98_{\pm 0.06}}$ | $1.15_{\pm 0.07}$ | $532.34_{\pm 7.78}$ | $1.83_{\pm 0.08}$ | $1.55_{\pm 0.03}$ |
| $\text{PROSE}_{\text{Full+ICL}}$ | $6719.98_{\pm 1164.23}$ | $1.77_{\pm 0.10}$ | $\mathbf{1.46_{\pm 0.05}}$ | $7297.62_{\pm 980.54}$ | $1.64_{\pm 0.05}$ | $\mathbf{1.67_{\pm 0.04}}$ |

Table 7: `GPT-4o` + PROSE's performance on the two writing tasks measured by the correctness of inferred preferences (Pref. Sim.) and preference compliance (PPCM) compared against no-preference conditioning (NPC), true preference generation (Oracle), in-context learning (ICL), CIPHER (Gao et al., 2024), and ablations over PROSE's components. Results are the mean and standard error across five seeds. Best non-Oracle results per task are bolded.

## C.3. PRELUDE Results

Results on PRELUDE (Gao et al., 2024) for PROSE and baselines: a no-preference conditioning (NPC), an oracle preference baseline, in-context learning (ICL), CIPHER-1, and CIPHER-5 (Gao et al., 2024) (Table 8). To directly evaluate the ability to infer preferences, we provide all models with ground-truth knowledge of the source of the documents. On the summarization task, PROSE outperforms all baselines on action/generation quality. On the email writing task, PROSE outperforms all baselines on the PPCM metric, but slightly underperforms CIPHER-1 on the poorly correlated Levenshtein distance metric (see Section 4-**Metric Correlation** for issues with Levenshtein distance).

Results in this table further support issues with the current preference-quality metrics. In the email writing task, the no-learning baseline (which always uses an empty preference), has a higher accuracy than any learning method, which may be due to the significant overlap between preference sets in the task. Further, in both tasks, the highest preference-quality scores do not lead to the highest action-quality scores. We encourage future work to look into alternative preference-quality metrics.

We lastly note that PRELUDE has substantially smaller range between the no-preference conditioned (NPC) and oracle preference baselines relative to PLUME. On PPCM, PRELUDE has a range 2.45 and 0.62 for summarization and email writing respectively, while PLUME has ranges of 3.17 and 2.91 for the two tasks. This further supports PLUME as the primary evaluation environment.

| Summarization | | | | |
|---|---|---|---|---|
| Method | Accuracy | BScore | Levenshtein | PPCM |
| No Learning Baselines | | | | |
| NPC | $0.20_{\pm 0.00}$ | $-0.43_{\pm 0.00}$ | $107.80_{\pm 6.04}$ | $-0.74_{\pm 0.10}$ |
| Oracle | $1.00_{\pm 0.00}$ | $1.00_{\pm 0.00}$ | $1.08_{\pm 1.48}$ | $1.62_{\pm 0.09}$ |
| Learning Baselines | | | | |
| ICL | $0.20_{\pm 0.00}$ | $-0.43_{\pm 0.00}$ | $104.24_{\pm 8.85}$ | $-0.71_{\pm 0.19}$ |
| CIPHER-1 | $0.61_{\pm 0.06}$ | $0.13_{\pm 0.03}$ | $48.01_{\pm 10.76}$ | $0.74_{\pm 0.24}$ |
| CIPHER-5 | $0.46_{\pm 0.02}$ | $0.02_{\pm 0.01}$ | $33.86_{\pm 19.89}$ | $0.82_{\pm 0.50}$ |
| PROSE | $0.68_{\pm 0.12}$ | $0.01_{\pm 0.03}$ | $9.30_{\pm 8.70}$ | $1.18_{\pm 0.16}$ |

| Emails | | | | |
|---|---|---|---|---|
| Method | Accuracy | BScore | Levenshtein | PPCM |
| No Learning Baselines | | | | |
| NPC | $0.25_{\pm 0.00}$ | $-0.37_{\pm 0.00}$ | $48.12_{\pm 11.44}$ | $0.86_{\pm 0.08}$ |
| Oracle | $1.00_{\pm 0.00}$ | $1.00_{\pm 0.00}$ | $1.02_{\pm 2.03}$ | $1.57_{\pm 0.13}$ |
| Learning Baselines | | | | |
| ICL | $0.25_{\pm 0.00}$ | $-0.37_{\pm 0.00}$ | $53.38_{\pm 13.46}$ | $0.87_{\pm 0.13}$ |
| CIPHER-1 | $0.03_{\pm 0.06}$ | $-0.16_{\pm 0.04}$ | $12.06_{\pm 15.04}$ | $1.04_{\pm 0.12}$ |
| CIPHER-5 | $0.25_{\pm 0.00}$ | $-0.08_{\pm 0.04}$ | $13.34_{\pm 9.57}$ | $1.09_{\pm 0.08}$ |
| PROSE | $0.01_{\pm 0.03}$ | $-0.22_{\pm 0.02}$ | $14.05_{\pm 6.14}$ | $1.10_{\pm 0.06}$ |

Table 8: **PRELUDE Results**. PROSE's ability to infer the correct preference set and generation quality across the two PRELUDE tasks compared against a no-preference conditioning baseline (NPC), a method with access to the true preferences (Oracle), in-context learning (ICL), and CIPHER (Gao et al., 2024). Results are reported as the mean and standard error across five seeds. Accuracy and Bscore (BERTScore) (Zhang* et al., 2020) are preference-quality metrics, while Levenshtein distance and PPCM (per preference-component match) are action-quality metrics.

## C.4. Preference Inference and Conditioning Performance by Number of User Samples

In Figure 6 we show the impact of the number of samples for a given user according to the measures for inferred-preference and generation quality metrics.

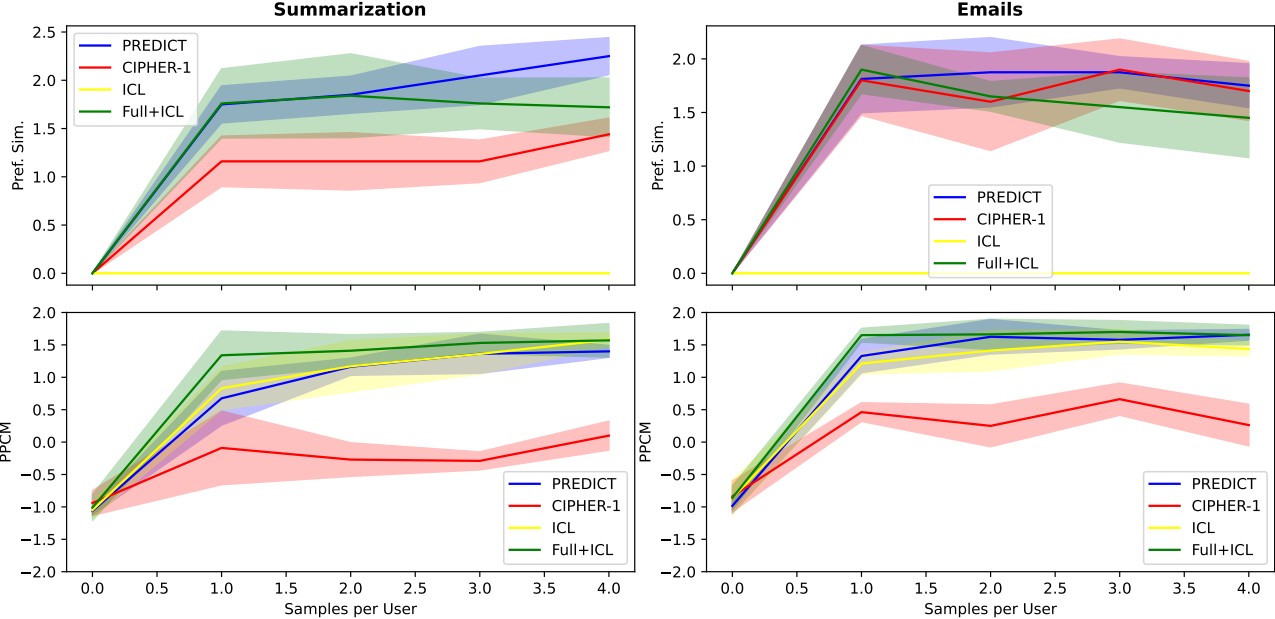

Figure 6: Performance for PROSE, CIPHER-1, in-context learning (ICL), and PROSE+ICL given different numbers of user samples to learn from. Mean and standard error over five seeds for preference quality (Pref. Sim.) and preference-conditioned generation quality (PPCM). `GPT-4o` is the LLM.

# D. PRELUDE vs. PLUME Preference Sets

The preference sets used for each document source and environment (PRELUDE vs. PLUME) are given in Table 9.

| Document Source | Task Version | User Preferences |
|---|---|---|
| | | Summarization |
| News Articles | PRELUDE | interactive, playful language, positive, short sentences, storytelling, style targeted to young children |
| | PLUME | adopt a step-by-step structure, include a simile, use ampersands (&) instead of "and"s, write in the style of a children's book |
| Chat Forum Posts | PRELUDE | brief, immersive, invoke personal reflection, second person narrative, show emotions |
| | PLUME | adopt a header and sub-header structure, include rhetorical questions, use ALLCAPS to emphasize words, write in the style of a tweet |
| Encylopedia Pages | PRELUDE | brief, bullet points, parallel structure |
| | PLUME | adopt a rhyming structure, include modern slang, use semicolons (;) when possible, write in the style of a screenplay |
| Paper Abstract | PRELUDE | inquisitive, simple English, skillful foreshadowing, tweet style, with emojis |
| | PLUME | adopt a question-answering style structure, include personifications, use archaic language, write in the style of a podcast |
| Movie Review | PRELUDE | question answering style |
| | PLUME | adopt a stream-of-consciousness structure, include onomatopoeias, use imagery, write in the style of old timey radio |
| | | Email Writing |
| Personal Problem | PRELUDE | conversational, informal, no closing |
| | PLUME | be intensely emotional, include alliterations, use a formal tone, write in a second person narrative |
| Paper Review | PRELUDE | call to action, casual tone, clear, positive |
| | PLUME | be sharply critical, include several short and punchy sentences, use parenthetical asides, write using assertive expressions |
| Paper Tweet | PRELUDE | engaging, personalized, professional tone, thankful closing |
| | PLUME | be blatantly sarcastic, include hyperboles, use an informal tone, write in a third person perspective |
| Paper Summary | PRELUDE | professional greeting and closing, respectful, straight to the points, structured |
| | PLUME | be highly inquisitive, include several long and flowing sentences, use emojis, write using conditional expressions |

Table 9: The user preferences for each assistive writing task (summarization vs. email writing), task topic, and benchmark version.

# E. Illustrative Examples of Issues with PRELUDE

## E.1. Levenshtein Distance and Multiple Generations

We present here an inherent issue with Levenshtein distance. For all three generations, we prompt gpt-4o to summarize the encyclopedia article on hotdogs. The first two generations (A. and B.) use identical prompts, in which we ask it to use the preference: "use a haiku". In the third generation, we ask it to use the preference "use a single short sentence". We then calculate the Levenshtein distance between each pair, reported in Table 10. Notably, the Levenshtein distance and length normalized Levenshtein distance, in which a lower score represents higher similarity, would both indicate that generation A and C are more similar to generation A and B, even though generation A and B share the same exact same prompt while generation C uses a entirely different preference. BERTScore does a better job of identifying the most similar texts, however the PPCM scores, which compares writing directly to the true preferences, clearly labels both Generation A. and Generation B. as "strongly exhibiting" the preference "use a haiku", and Generation C. as "strongly contradicting" the preference.

Generation A.

> A sausage in bun,
> Toppings vary near and far,
> Hot dog, rich in roots.

Generation B.

> Grilled or steamed sausage,
> In a bun with cheese or spice,
> Street food's warm embrace.

Generation C.

> A hot dog is a sausage in a bun with various toppings such as mustard, ketchup, and relish

| Pairing | Levenshtein Dist. | Norm. Levenshtein Dist. | BERTScore | PPCM per generation |
|---------|-------------------|-------------------------|-----------|---------------------|
| A-B     | 17                | 0.85                    | 0.27      | (2, 2)              |
| A-C     | 16                | 0.76                    | 0.20      | (2, -2)             |
| B-C     | 17                | 0.81                    | 0.10      | (2, -2)             |

Table 10: Levenshtein distance between each pair of generations.

### E.2. Editing Influences the User

In this next example, we demonstrate how the editing procedure can influence the output of the user. We again ask an LLM to summarize the encyclopedia article on hot dog. Let's assume, an inferring agent has incorrectly inferred the preference "question answering style". Using this preference, the preference-conditioned agent writes:

> What is a hot dog?
>
> A hot dog is a dish made of a sausage (typically a wiener or frankfurter) served in a partially sliced bun. It can refer to the sausage itself, which is grilled, steamed, or boiled.
>
> How is a hot dog typically prepared?
>
> Hot dog preparation varies, but common condiments include mustard, ketchup, relish, and onions. Some toppings can also include cheese sauce, sauerkraut, chili, bacon, or jalapeños.
>
> Where did hot dogs originate?
>
> The sausages used in hot dogs were culturally imported from Germany, and the dish became popular in the U.S. as a working-class street food. Over time, hot dogs became closely associated with American culture, particularly baseball.
>
> Where are hot dogs most popular?
>
> While they are especially associated with New York City, hot dogs became widespread across the United States throughout the 20th century, becoming an integral part of American cuisine.

We ask the user if the above is satisfactory for it's true preference of: "style targeted to young children". It does not, so we then ask the user to edit the above for it true preference "style targeted to young children", and the user writes:

> What is a hot dog?
>
> A hot dog is a yummy snack made of a sausage (like a wiener or frankfurter) inside a soft bun. Sometimes people call the sausage itself a hot dog, and it can be cooked by grilling, steaming, or boiling.
>
> How is a hot dog usually made?
>
> Hot dogs can be made in different ways, but they often have yummy toppings like mustard, ketchup, relish, and onions. Some people like to add even more fun stuff like cheese, sauerkraut, chili, bacon, or spicy jalapeños.
>
> Where did hot dogs come from?
>
> The sausages in hot dogs came from Germany, and a long time ago, they became super popular in the U.S. as a quick and tasty street food for busy people. Now, hot dogs are a big part of American culture, especially at baseball games!
>
> Where are hot dogs the most popular?
>
> Even though hot dogs are popular all over, they are especially loved in New York. Over the years, hot dogs have become a favorite food all across the United States!

While the above summary does correctly follow the true preference, it is still formatted in a question answering style, which is not part of the user's preferences. I.e., the preference-inferring agent's error has directly impacted the user's writing. This is another reason why PLUME removes the editing procedure.

For reference, if the user is directly asked to summmarize the encyclopedia article on hot dogs using it's true preference of "style targeted to young children", it outputs:

> A hot dog is a yummy food that's made with a sausage, like a wiener or frankfurter, inside a soft bun. You can cook the sausage in different ways, like grilling or boiling. People love to put tasty toppings on their hot dogs, such as ketchup, mustard, cheese, and even onions!
>
> Hot dogs came from Germany, but they became super popular in the US, especially at baseball games. Now, you can find hot dogs all over the country, and in some places, like Chicago, people have special ways of making them!

# F. Prompts

### F.1. Preference Inference and Preference-Conditioned Agent Prompts

The prompts used by PROSE for writing sample generation and task completion in the PLUME environment are in Appendix Figure 7. The prompts used by PROSE to infer the user preferences from user examples are provided in Appendix Figure 8 for the PLUME environment.

---

**System Prompt**

You are an experienced writer. Adapt your writing to heavily emphasize the provided preferences.

**User Prompt**

You have the following preferences: `<preference_description>`

Using these preferences, write a short {`summary` | `email`} about {`this` | `these`} {`article` | `notes`}:

[START OF {`ARTICLE` | `NOTES`}]
`<task_content>`
[END OF {`ARTICLE` | `NOTES`}]

Encapsulate the {`summary` | `email`} in triple quotes
"""
`<{summary | email}>`
"""

---

Figure 7: LLM prompts for the **preference-conditioned agent** and for **task completion** on the **PLUME's** summarization and e-mail writing tasks. The system prompt is prepended to the user prompt following the LLM's chat template. "{...|...}" means that of the two options is selected based on the task and "`<...>`" indicates that the text is formatted from a variable. `inferred_preference_i` refers to one of the inferred user preferences.

---

**System Prompt**

A user is completing writing tasks. The user has an underlying set of preferences that explains why they write the way they do.

**User Prompt**

**Aggregation Task**

We are tasked to curate a prompt to guide a specific style of writing. We currently have the following list of preferences related to writing styles:
[`<preference_description`,..., `<inferred_preference_l>`]
Unfortunately, these preferences may overlap or contain redundancies. Please review the list and condense it by combining similar or overlapping preferences, ensuring that the distinct intent behind each one remains clear so that a writer can easily follow them. Ensure the condensed list is concise, non-redundant, and preserves the original level of specificity. When applicable, preserve the exact wording. Return the revised preferences in the same format as the original list.

**Inference Task**

We received a new task. The task is to {`summarize` | `write an email about`} the following:
`<article` | `notes>`

We have previously identified the following preferences: `<preference_description>`
Based on these preferences, we wrote this {`summary` | `email`}:
`<assistant_output>`

However, this differs from the user's {`summary` | `email`}. The user wrote this {`summary` | `email`}:
`<user_output>`

Refine the list of preferences by adding, removing, or updating preferences in order to better imitate the user.

While refining the preference set, you should:
- Identify and reason about differences between our writing and the user's writing.
- Consider writing traits from distinct quirks to broader stylistic tendencies.
- Provide a concise set of preferences in the imperative form.
- Be precise; make the fewest possible changes to the preference set.
- Do not qualify, dilute, or soften existing preferences.
- Refine only the preferences if a clear difference exists. Otherwise, preserve the current preferences.

Provide a concise set of specific preferences in the imperative form. After reasoning, , output the refined set of preferences on a single new line and prefaced with "Preferences:".

Figure 8: LLM prompts for **preference inference** on **PLUME's** summarization and e-mail writing tasks. The system prompt is prepended to each user prompt following the LLM's chat template. "{...|...}" means that of the two options is selected based on the task and "`<...>`" indicates that the text is formatted from a variable. `user_output` refers to how the user completes the task, `assistant_output` how the assistant completes the task, and `inferred_preference_i` to one of the inferred user preferences. Continued on next page.

---

**User Prompt**

**Preference Breakdown Task**

You inferred the following preference string:
`<inferred_preference_description>`
Format this preference into a concise set of preferences. Format the final set of preferences as a JSON list on a single line and prefaced with "Preferences:". Each element in the JSON list should be a string.The final output should look like:
Preferences: [<preference 1>,..., <preference i>, ...]

**Consistency Verification Task**

Validate the following preference: "`<inferred_preference_i>`" against the following writing:

`<user_output>`

Does the writing confirm or contradict the preference? Select one of the following: strongly confirms the preference, somewhat confirms the preference, is neutral toward the preference, somewhat contradicts the preference, strongly contradicts the preference. Your final decision should be output on a separate line prefaced with "Verdict:".

Figure 8: LLM prompts for **preference inference** on the **PLUME's** summarization and e-mail writing tasks. The system prompt is prepended to each user prompt following the LLM's chat template. "{...|...}" means that of the two options is selected based on the task and "`<...>`" indicates that the text is formatted from a variable. `user_output` refers to how the user completes the task, `assistant_output` how the assistant completes the task, and `inferred_preference_i` to one of the inferred user preferences.

## F.2. Synthetic Human Prompts

The prompts used to have GPT-4o play the role of our synthetic human for PROSE are given in Appendix Figure 9. The "human" is instructed to complete the task in the same way as the preference-conditioned agent when completing the writing tasks (see Appendix Figure 7).

---

**System Prompt**

You are an experienced writer. Adapt your writing to heavily emphasize the provided preferences.

**User Prompt**

You have the following preferences: <ground_truth_preference_description>

Using these preferences, write a short {summary | email} about {this | these} {article | notes}:

[START OF {ARTICLE | NOTES}]
<task_content>
[END OF {ARTICLE | NOTES}]

Encapsulate the {summary | email} in triple quotes
"""
<{summary | email}>
"""

---

Figure 9: LLM prompts for the **synthetic human** on the **PLUME's** summarization and e-mail writing tasks. The system prompt is prepended to the user prompt following the LLM's chat template. "{...|...}" means that of the two options is selected based on the task and "<...>" indicates that the text is formatted from a variable. inferred_preference_i refers to one of the inferred user preferences.

### F.3. Preference-Conditioned Agent Baseline Prompts

The prompts used in the no-preference baseline are in Appendix Figure 10 and for the in-context learning baseline are in Appendix Figure 11. For the in-context learning baseline, the number of examples $l$ matches the number of examples used when coalescing prevoiusly inferred prompts (see Appendix Figure 8).

---

**System Prompt**

You are an experienced writer.

**User Prompt**

Write a short {`summary` | `email`} about {`this` | `these`} {`article` | `notes`}:

[START OF {`ARTICLE` | `NOTES`}]
`<task_content>`
[END OF {`ARTICLE` | `NOTES`}]

---

Figure 10: LLM prompts for the **no preference baseline** in the **PLUME environment**. The system prompt is prepended to the user prompt following the LLM's chat template. "`<...>`" indicates that the text is formatted from a variable. `task_content` refers to the content of either the article to be summarized or the notes to include in the email, depending on the sub-task.

**System Prompt**

You are an experienced writer. Adapt your writing to heavily emphasize the provided preferences.

**User Prompt**

You have previously observed the following examples:

Example 0:
{Article | Notes}:
[START OF {ARTICLE | NOTES}]
`<task_content>`
[END OF {ARTICLE | NOTES}]

{Article | Notes}:
"""""
`<completion_0>`
"""""

.
.
.

Example $l$:
{Article | Notes}:
[START OF {ARTICLE | NOTES}]
`<task_content>`
[END OF {ARTICLE | NOTES}]

{Article | Notes}:
"""""
`<completion_l>`
"""""

Using the same style as these examples, write a short {summary | email} about {this | these} {article | notes}:

[START OF {ARTICLE | NOTES}]
`<task_content>`
[END OF {ARTICLE | NOTES}]

Encapsulate the {summary | email} in triple quotes
"""""
`<{summary | email}>`
"""""

Figure 11: LLM prompts for the **in-context learning baseline** in the **PLUME environment**. The system prompt is prepended to the user prompt following the LLM's chat template. "`<...>`" indicates that the text is formatted from a variable, and completion_$l$ refers to an example completion provided for in-context learning. task_content refers to the content of either the article to be summarized or the notes to include in the email, depending on the sub-task.

## F.4. Qualitative Verification Consistency Examples

| Preference components after refinement | `Most relevant true preference` OR *Notes* |
|---|---|
| Write in a whimsical, playful, and narrative style using vivid and childlike imagery | `Write in the style of a children's book` |
| Maintain a hopeful, conversational, and informal tone | `Write in the style of a children's book` |
| ~~Mention geographical context early in a simple manner~~ | *Overfit to a specific example* |
| Focus on the sequence of events with explicitly numbered steps labeled as 'first,' 'next,' 'then,' 'after that,' and 'finally' | `Adopt a step-by-step structure` |
| Use simple and metaphorical language for emotional aspects | `Include a simile` |
| ~~Include personal details about characters~~ | *Overfit to a specific example* |
| Use ampersands for conjunctions | `Use ampersands (&) instead of ''and''s` |
| Conclude with a practical reminder or lesson emphasizing support and teamwork | `Write in the style of a children's book` |
| ~~Use rhetorical questions sparingly~~ | *Discarded because they were used more than sparingly* |
| Capitalize key traits/actions for emphasis | `Use ALLCAPS to emphasize certain words` |
| Use third-person perspective | `Adopt a third-person narrative` |
| Use emojis strategically | `Write in the style of a tweet` |
| Use informal and playful language | `Write in the style of a tweet` |
| Use hashtags to encapsulate themes | `Write in the style of a tweet` |
| ~~Minimize emotionally charged phrases~~ | *Irrelevant* |
| Use direct questions | `Include rhetorical questions` |
| ~~Be concise and direct~~ | *Irrelevant* |
| Focus on emotional impact and highlight key themes | *Incorrect, but related to the content of many chat forum posts* |
| Highlight internal conflict | *Incorrect, but related to the content of many chat forum posts* |
| use direct address with a casual and contemporary tone | `Include modern slang` |
| include character interactions with informal dialogue and consistent rhyming couplets | `Adopt a rhyming structure` |
| employ a simplified screenplay format focusing on dialogue and voiceover | `write in the style of a screenplay` |
| incorporate slang and playful language | `Include modern slang` |
| highlight excitement and stakes dynamically | *Loosely related to screenplay* |
| use thematic transitions between sections | `Write in the style of a screenplay` |
| ~~include metric prefixes explicitly~~ | *Overfit to a specific example* |

Table 11: Qualitative examples of Verification. ~~Strikethrough~~ indicates the preference component was pruned. Verification successfully removes overfit of irrelevant preferences. On occasion, it discards relevant, but misqualified components.

## F.5. Qualitative Iterative Refinement Examples

| Refinement step | Inferred preference descriptions |
| --- | --- |
| True preferences | adopt a third person narrative, include rhetorical questions, use ALLCAPS to emphasize certain words, write in the style of a tweet |
| 1 | **Use rhetorical questions, capitalize for emphasis, be concise, include symbols or emojis, focus on emotional impact.** |
| 2 | Use rhetorical questions, capitalize for emphasis, be concise, include symbols or emojis, focus on emotional impact, **use hashtags, use symbols for brevity.** |
| 3 | Use rhetorical **questions strategically**, capitalize for emphasis, be concise, **limit** symbols or emojis, focus on emotional impact, **use 1-2 hashtags**, use symbols for brevity. |
| 4 | Use rhetorical questions strategically, capitalize for emphasis, be concise, limit **symbols**, focus on emotional impact, use 1-2 hashtags, use symbols for brevity, **incorporate emojis for emphasis.** |
| 5 | Use rhetorical questions strategically, capitalize for emphasis, be concise, **use "&" for brevity**, focus on emotional impact, use 1-2 hashtags, **use fewer emojis for emphasis, highlight key themes.** |
| True preferences | adopt a question-answering style structure, include personifications, use archaic language, write in the style of a podcast |
| 1 | **Use a poetic and narrative style with vivid imagery and metaphor; employ archaic language and a conversational tone; structure writing like a narrative or script; directly address the audience to enhance engagement.** |
| 2 | Use a **podcast or broadcast format** with vivid storytelling imagery and metaphor; employ **slightly modern** archaic language and a conversational tone; structure writing **as a continuous narrative**; directly address the audience to enhance engagement. |
| 3 | Use a podcast or broadcast format with **named episodes**; employ **consistently** archaic language **with varied vocabulary**; **use thematic and specific metaphors; enhance audience engagement with direct questions and conversational elements**; structure writing **with a clear narrative arc and defined sections**. |
| 4 | Use a podcast or broadcast format with **creatively thematic episode titles**; employ consistently archaic **and poetic** language with varied vocabulary; **use vivid and personified metaphors**; enhance audience engagement with **rhetorical style and subtle questions**; structure writing with a clear narrative arc and **poetic conclusions.** |
| 5 | Use a podcast or broadcast format with creatively thematic episode **titles evoking transformation and mystery**; employ **consistently archaic** language throughout; use vivid and personified metaphors; enhance audience engagement with **frequent** rhetorical questions; structure writing with a clear narrative arc and **conclude with harmony and enlightenment.** |
| True preferences | adopt a second person narrative, include onomatopoeias, use imagery, write in the style of old timey radio |
| 1 | **Use vivid imagery and metaphor, adopt a narrative style, address the reader directly, and create an immersive experience.** |
| 2 | Use vivid imagery and metaphor, adopt a narrative style, address the reader directly **with a conversational and auditory tone**, create an immersive **and nostalgic** experience. |
| 3 | Use vivid imagery and metaphor, adopt a narrative style, address the reader directly with a conversational and auditory tone, create an immersive **experience with a focus on auditory imagery and live storytelling.** |
| 4 | Use vivid **auditory** imagery and metaphor, adopt a narrative **style reminiscent of a radio broadcast**, address the reader directly with a conversational and auditory tone, create an immersive experience with a focus on **auditory** storytelling. |
| 5 | Use vivid auditory imagery and metaphor **with a nostalgic and whimsical tone**, adopt a narrative style reminiscent of a classic radio broadcast, address the reader directly with a conversational and auditory tone, create an immersive experience with a focus on auditory storytelling. |

Table 12: Qualitative examples of Iterative Refinement. Bold indicates modifications performed by the refinement step. Note: the true preferences are for reference only, the initial refinement step is conditioned on an empty preference description.

