# OpenReview forum: "Aligning LLMs by Predicting Preferences from User Writing Samples"
_ICML.cc/2025/Conference — ICML 2025 poster_

### Official Review · Reviewer_CfWS · 2025-03-10

**Overall Recommendation:** 3

**Summary:**

This paper introduces PROSE, a method for inferring precise and personalized user preferences to enhance LLM-based writing agents. The approach employs iterative refinement and cross-sample verification to generate more accurate preference descriptions compared to existing techniques. Alongside PROSE, the authors introduce PLUME, a new benchmark specifically designed for evaluating preference learning in assistive writing tasks. Through experiments on summarization and email writing tasks across multiple LLMs, PROSE demonstrates a significant improvement over the state-of-the-art CIPHER method for preference inference. Additionally, the author demonstrate that combining PROSE with in-context learning (ICL) lead to further performance gains.

### update after rebuttal
I appreciate the authors' detailed responses and the addition of the human evaluation. I will maintain my current score.

**Claims And Evidence:**

The paper provides convincing evidence through quantitative experiments, with tables 1-2 and figures 2-3 showing consistent performance improvements across different LLMs and writing tasks. Additionally, the ablation studies and analysis provide convincing evidence of the iterative refinement component's effectiveness. Finally, fig. 2 and tab.2 show that PROSE combined with ICL provide additional benefit and this is consistent across different models and taks.

The paper makes broad claims about generalizability but this would require evaluation on more diverse writing tasks and not only summarization and email writing tasks. Also, the paper states that consistency verification "prunes irrelevant preferences" but provides limited quantitative evidence on how effectively it identifies truly irrelevant vs. simply uncommon preferences. Finally,the evaluation relies heavily on automated metrics. While these are the standard, an additional human evaluation would strengthen claims about real-world preference alignment and writing quality improvements.

**Essential References Not Discussed:**

The paper does not discuss parallels with these two works that appear to be somewhat relevant. "Show, Don't Tell: Aligning Language Models with Demonstrated Feedback" Shaikh et al published in ICLR 2025. This is very recent but the arxiv version is from June 2024.
This work introduces DITTO, a method that directly aligns language models to user-demonstrated behaviors using a very small number of demonstrations as feedback. Like PROSE, DITTO focuses on learning from user demonstrations rather than explicit feedback, but employs online imitation learning principles, providing an alternative theoretical framing. DITTO specifically addresses personalization in emails and other writing tasks that directly overlap with PROSE's evaluation domains. DITTO reports substantial improvements (19% points on average) over few-shot prompting and other methods. PROSE should have compared against or at least discussed this performance benchmark.

Another work is "Unsupervised Human Preference Learning" by Shashidhar et al, 2024 from ACL 2024. This paper proposes using small parameter models as preference agents to generate natural language rules that guide larger pre-trained models. Like PROSE, this work leverage natural language descriptions of preferences to guide language model outputs, rather than direct parameter updates and aims to achieve personalization without requiring full model fine-tuning, highlighting efficient approaches to customization.

**Experimental Designs Or Analyses:**

The experimental design is generally sound and appropriate for evaluating the proposed method. The PLUME benchmark provides a needed framework for evaluating preference learning. The most significant validity concern is the reliance on synthetic preferences and LLM-as-judge evaluation, which are practical compromises but may not fully represent real human preference dynamics. The paper acknowledges these limitations appropriately, which strengthens the overall scientific integrity of the work.

**Methods And Evaluation Criteria:**

I find that the methods and evaluation criteria/metrics are well aligned with the problem of inferring anf applying user wiriting rpeferences. The method addresses limitations of previous approaches, and evaluates them in a systematic manner. Additionally, the creation of PLUME as a specialized benchmark for this task is particularly valuable, though expanding task diversity and incorporating more human evaluation would further strengthen the assessment of the method's practical utility.

**Other Comments Or Suggestions:**

see questions.

**Other Strengths And Weaknesses:**

__Strengths__:
- the idea of combining iterative refinement and coss-sample verification lead to perform preference learning from a 1-shot process to a progressive refinement problem
- the paper is well written and easy to follow
- comprehensive evaluation and PLUME is a valuable contribution
- the 33%improvement over CIPHER demonstrate the validity of this approach

__Weaknesses__:
- there is a limited task selection, only summariation and email writing. this leave the open question of how well PROSE generalizes to other types of texts.
- The evaluation relies entirely on automated metrics without human assessments of quality or preference alignment, leaving open questions. Although the need for a full-scale human evaluation is listed in the limitation sections, a small sample submitted to human evaluation could help better evaluate the claims of the paper.

**Questions For Authors:**

- You mention that sorting preference components by length before aggregation led to an 11% performance drop, which is an intriguing finding. Have you explored other ordering strategies for preference components, and do you have insights into why component ordering has such a significant impact on performance?

- How does PROSE handle cases where user demonstrations contain genuinely conflicting preference signals?
-if a user writing preferences evolve over time or vary depending on the context or the topic, how might PROSE be adapted to handle preference drift or contextual preference variations?

**Relation To Broader Scientific Literature:**

Besides the evident ones, e.g. PROSE extends CIPHER in inferring preferences directly from demonstrations rather than explicit feedback connecting to the literature for learning from implicit signals, this work is connected to the body of work on personalization through preference modeling, self-refinement in LLMs, reflection in LLMs and somehow to writing style transfer.

**Theoretical Claims:**

The paper does not present theoretical claims due to the empirical nature. The paper describes the iterative refinement process that continues until either the candidate solutions exactly match the user demonstrations or a maximum number of iterations is reached, but does not make theoretical claims about convergence guarantees.

---

> ### Author Rebuttal · Authors · 2025-03-31
>
> Thank you for your feedback and questions. We appreciate that you find PLUME and our comprehensive evaluation to be valuable contributions along with our learning about the importance of iterative refinement.
>
> **Response to questions**
>
> *Re your question about sorting preference components:* We also tried sorting alphabetically and observed similar performance differences. The impact of ordering has been previously identified in work on multiple choice QA tasks [1], and we think this is related. Additionally, we hypothesize the LLM is doing some level of prompt engineering on itself when it infers the user’s preferences, and that the preference ordering is part of this. As we know from the prompt engineering literature, small changes to the prompts can have a large impact on LLM generations.
>
> *Re your question about conflicting preferences, context sensitivity, and changing preferences:* The issues of contextual preference variations and preference evolution over time can be solved with the example retrieval step. By retrieving the most contextually relevant and temporally local examples from memory, PROSE should be able to handle preference drift and context variations. This then largely becomes a problem of constructing a sufficiently detailed context. However, for this paper, we identify the retrieval step as an orthogonal issue to inferring preference descriptions, so do not tackle it.
>
> **Response to "weaknesses”**
>
> *Re limited task selection:* Our task selection is based on those used in previous work and specifically targeting “write on my behalf” tasks. The other types of tasks that exist in the LLM literature, such as dialogue or question answering, do not fall under “write on my behalf”. The “write on my behalf” task is a common one people ask LLMs to do, which motivated our selection of it. The approach of learning from demonstrations we take with PROSE is not relevant to other tasks such as question-answering or dialogue because in those other tasks the user is asking the LLM to complete tasks they have not done.
>
> *Re human evaluation:* Thank you for this feedback. We have run a human evaluation with 16participants (3 are ML researchers; 9 women and 7 men; age in [19, 58]). Participants completed a within subjects AB test comparing PLUME+ICL generations to ICL generations and PLUME+ICL generations to CIPHER generations.
>
> Participants evaluated LLM generations for two different preferences for the email task and for the summarization task for a total of 20 survey items per method comparison. We used the responses to compute a win rate for PLUME+ICL compared to each of ICL and CIPHER. The only difference between the two comparisons is the LLM generations compared.
>
> For PLUME+ICL versus ICL, we see an average win rate of 69.4%. For PLUME+ICL versus CIPHER, we see an average win rate of 91.8%. The human evaluation results support our synthetic evaluation results.
>
> **Response to missing related work**
>
> Thank you for the feedback that our related work would benefit from discussing DITTO and "Unsupervised Human Preference Learning". We agree there are important parallels to discuss with both papers. We have referenced DITTO in the introduction as another method for learning from demonstrations, and agree it should also be mentioned in the Related Work section. We will additionally reference and discuss "Unsupervised Human Preference Learning" in future revisions of the paper, as we believe their method and PROSE are complementary; PROSE would likely benefit from leveraging a model that is directly trained to infer preferences, and, as their method also revolves around reducing the delta between generic and user responses, their M_L model would likely benefit from the iterative refinement and consistency verification proposed in PROSE.
>
>
>
> [1] Pezeshkpour, P., & Hruschka, E. (2023). Large language models sensitivity to the order of options in multiple-choice questions. arXiv preprint arXiv:2308.11483.

---

### Official Review · Reviewer_Wbtm · 2025-03-14

**Overall Recommendation:** 3

**Summary:**

The paper introduces PROSE (Preference Reasoning by Observing and Synthesizing Examples), a method for aligning large language models (LLMs) with user preferences inferred from writing samples. It improves upon previous sota/baselines on various aspects, proposes new metrics and preference frameworks, and provides a new dataset for writing preference alignment.

**Claims And Evidence:**

The main claims of the paper and their supporting evidence are:

- PROSE produces more precise preference descriptions than CIPHER – This is supported by experiments on PLUME, showing a 33% improvement in preference alignment.

- Iterative refinement improves alignment between LLM generations and user preferences – Ablation studies demonstrate that increasing the number of refinement steps improves performance by 14.8%.

- Consistency verification enhances robustness – Though its effect is smaller (1.5–1.7% improvement), the verification step helps filter irrelevant or overfit preferences.

- PROSE and ICL are complementary – Combining PROSE with ICL leads improvement.

- improved dataset PLUME - The authors analyze PRELUDE’s limitations (e.g., weak correlation between inferred preferences and generation quality) and show that PLUME provides more consistent evaluation metrics.

These claims are well-supported by empirical results. The arguments are well-written and limitations are discussed.

**Essential References Not Discussed:**

N/A

**Experimental Designs Or Analyses:**

No notes, well done. I have a question about the setup (see question 3.)

**Methods And Evaluation Criteria:**

See question 1b & 2.

**Other Comments Or Suggestions:**

Typo: line 45 left "share share"

**Other Strengths And Weaknesses:**

Overall the paper is well-motivated and well-written. I believe the problem is an important one, the evaluation thorough, and this paper not only improves upon previous methods but also provides a better benchmark than PRELUDE. Graphs are nice and clear. The discussion on limitation addresses questions about computational costs.

For weakness, please see questions for authors section.

**Questions For Authors:**

(1a). I was confused about table 1 until I read through page 5. Please write out all the acronyms out in the caption, and explain the main message of table 1 (Preference similarity is a better metric?)  Also, please provide the formula of how you calculated the correlation for checking statistical validity.

(1b). The point of the R correlation test is also unclear to me - since you are using the PPCM metric already, why do you want another metic that highly correlates with it? Since they are testing different thing (preference vs. generation quality), don't you want them to be as orthogonal as possible? (i know in reality a good response usually scores high on both, so empirically it's hard to show. But I wonder the reasoning behind the test.)

2. About preference set specification on page 5 bottom of left column. Is equal # of preference & the same structure necessarily a good thing? Wouldn't that limit the generality of the benchmark?

3. Would PROSE still be effective if applied to real users instead of synthetic GPT-4o proxies? Have you considered testing with human participants to validate that inferred preferences align with subjective user expectations?

**Relation To Broader Scientific Literature:**

This work builds on previous efforts in preference learning for LLM alignment, particularly CIPHER (Gao et al., 2024) and in-context learning.

**Theoretical Claims:**

No theoretical claims.

---

> ### Author Rebuttal · Authors · 2025-03-31
>
> Thank you for your feedback and questions. We appreciate that you find the problem important, our evaluation thorough, and both our algorithm and benchmark improvements to be meaningful contributions.
>
> **Response to questions/”weaknesses”**
>
> **(1a)** Thank you for letting us know the table is confusing. We will expand the caption to include the information you recommend to specify each acronym and include the main message. We will also add the formula for how we calculated correlation: COV(X, Y) / (std(X) * std(Y)) where COV is covariance, std is standard deviation, X is the preference similarity scores and Y is the generation quality scores.
>
>
> **(1b)** Thank you for raising this question. We realize the issue is with how we present our evaluation metrics to imply that using PPCM is a given. Instead, we used the correlation test to select both PPCM and P.Sim. We tested a variety of different metrics, including those used by PRELUDE (which have a low correlation), prior to selecting PPCM (see Appendix C.1 Table 3). We selected the generation and preference inference metrics with the highest overall correlation.
>
> We want the measures of generation and preference inference quality to be correlated as correctly inferred preferences should lead to high-quality generations. For example, generating text conditioned on the ground truth preferences should have a higher generation score than when conditioning on the preferences for another user, no preference, or an incomplete preference set.
>
> We don’t want the two metrics to be orthogonal because they are measuring the quality of two different, but high correlated, things -- the inferred preferences versus the generation quality. We want to make sure both closely align with the user’s true preferences. Therefore, we are measuring both according to the same attribute.
>
>
> **(2)** Assigning each user the same number of preferences was a design decision we made so all users are similarly difficult. We did not want a scenario where a model could perform very well in aggregate by focusing only on those users with the fewest number of preferences to learn.
>
> By shared structure we mean each preference set has preferences over shared attributes for the writing. For example, format (e.g. screen play versus tweet), tone of voice (e.g. inquisitive versus sarcastic), or use of literary tools (e.g. rhyming versus alliteration). This design decision was also made to create user preference sets that were similarly challenging.
>
> Adding additional preferences and structures to the preference sets is straightforward to add and adapt the benchmark, which we encourage future researchers to do.
>
>
> **(3)**  We have run a human evaluation with 16 participants (3 are ML researchers; 9 women and 7 men; age in [19, 58]). Participants completed a within subjects AB test comparing PLUME+ICL generations to ICL generations and PLUME+ICL generations to CIPHER generations.
>
> Participants evaluated LLM generations for two different preferences for the email task and for the summarization task for a total of 20 survey items per method comparison. We used the responses to compute a win rate for PLUME+ICL compared to each of ICL and CIPHER. The only difference between the two comparisons is the LLM generations compared.
>
> For PLUME+ICL versus ICL, we see an average win rate of 69.4%. For PLUME+ICL versus CIPHER, we see an average win rate of 91.8%. The human evaluation results support our synthetic evaluation results.

---

### Official Review · Reviewer_wsxk · 2025-03-15

**Overall Recommendation:** 3

**Summary:**

This paper is an extension work following Gao, 2024. Agent alignment is achieved by conditioning on an inferred description of user preferences. Yet, existing methods often lead to generic descriptions that fail to capture the unique, individualized aspects of human preferences. To address this limitation, this paper introduces PROSE, a novel technique that enhances the precision of preference descriptions derived from user writing samples. To overcome the challenges in prelude, this paper proposes iterative refinement of preferences and verification framework.

**Claims And Evidence:**

Yes. The claims made on the experiment results and comparison are supported by evidence.

**Essential References Not Discussed:**

There is no literature missing directly related to the proposed framework. However, I would encourage to include more model alignment works in the paper. There are a few recent works focus on alignment without introducing additional trainable models. (Please see following)

[1] Li, Kenneth, et al. "Inference-time intervention: Eliciting truthful answers from a language model." Advances in Neural Information Processing Systems 36 (2023): 41451-41530.

[2] Turner, Alexander Matt, et al. "Activation addition: Steering language models without optimization." arXiv e-prints (2023): arXiv-2308.

**Experimental Designs Or Analyses:**

Yes. I have checked the experiment setting, benchmark and settings. The experiment design is similar to prelude benchmark and it makes sense.

**Methods And Evaluation Criteria:**

Yes. This work directly compares to the relevant prelude benchmark, and no learning, ICL.

**Other Comments Or Suggestions:**

I believe one challenge existing in prelude are not solved in the proposed PROSE framework: PROSE and prelude assumes the user's preference towards a topic is static. This makes the alignment easier than realistic case. However, in practice, the preference can depends on both topic and occasion, e.g, email to co-worker or manager? I would suggest use more complicated dataset, where the retrieval is not only based on the similarity between questions (or topic), but also based on occasions.

**Other Strengths And Weaknesses:**

Strength:

1. This paper directly points out the weakness of the prelude framework. I have tested the code base of prelude before, and discovered the similar problems (metric, editing process) in the prelude as the authors pointed out in the paper. Thus, I believe this work's improvement over prelude is important and makes sense.

2. The proposed methods to improve the prelude is effective and can be directly measured by the common evaluation methods.

3. The PROSE framework is efficient and effective compared to other alignment methods in literature, e.g, reward model training, fine-tuning.

Weakness:

1. The proposed PROSE framework heavily relies on prelude, which lacks novelty. The improvement over prelude basically focuses on introducing better evaluation metric, better process and prompting for editing, with additional verification process. However, there is **no fundamental change** compared to prelude. This makes the work seems to have limited contribution.

2. This work seems to claim the refinement procedure is novel (From line 106 to the end of the paragraph). In fact, the prelude framework also includes the refinement of inferred preferences. If you check the code of prelude, it includes a step to aggregate the currently inferred preference with the previous ones.

**Questions For Authors:**

1. As discussed in the previous part, is it able to extend the PROSE from static preference to varying preference depending on different occasions?

2. Besides the improvement in metric, prompting, and verification process, is there any **structural or fundamental** advantage over prelude? For example, can adapt to user preference in fewer rounds, or saving user editing effort.

I could consider changing scores if my concerns are addressed.

**Relation To Broader Scientific Literature:**

This work is based on the prelude framework [1]. This work improves the evaluation metric, editing process and adds a verification procedure. The improvement makes the user edit framework more efficient.

[1] Gao, Ge, et al. "Aligning llm agents by learning latent preference from user edits." arXiv preprint arXiv:2404.15269 (2024).

**Theoretical Claims:**

There is no theoretical claim.

---

> ### Author Rebuttal · Authors · 2025-03-31
>
> Thank you for your feedback and questions. We appreciate you finding PLUME to be effective and PROSE to be efficient and effective.
>
> **Response to “weaknesses”**
>
> (1) Our contributions are spread across our algorithmic developments in PROSE and our benchmark improvements in PLUME.
>
> In addition to improving the quality of PLUME’s evaluation protocols and prompting, we validated and improved the ground truth preferences to ensure the proxy humans are both sensitive to them when “writing” and when “evaluating”. If the proxy humans are not sensitive to the ground truth preferences then there is little signal in the training data and the results are meaningless. Additionally, we adapted PLUME to provide demonstrations as the data source from which to learn from, not edits.
>
> Our algorithmic contributions demonstrate the importance of validating inferred preferences across all relevant samples and iteratively refining the inferred preference description per sample. CIPHER (Gao et al., 2024) does neither.
>
> (2) Thank you for raising this concern and we will make sure the distinction is made clear in the paper. CIPHER includes a refinement step when observing a new sample, at which point they retrieve relevant samples, combine their preference, and refine this aggregated preference using the new sample. PROSE similarly does this aggregation step across retrieved examples. However, PROSE’s contribution is adding iterative refinement, where the preferences are refined several times using the same sample during preference description inference. This novel algorithmic contribution is distinct from CIPHER’s single pass method, and provides an 11% improvement on average. In addition to the iterative refinement step, PROSE’s contribution is the preference verification step. Reviewer CfWS describes these contributions as adapting the preferences learning process “from a 1-shot process to a progressive refinement problem.”
>
> **Response to questions**
>
> (1) Yes, this is handled in the example retrieval step, which is not a contribution of PROSE, therefore it is not discussed in detail. Given sufficient context about both the demonstrations and the current task, only those preferences relevant to the current context will be retrieved. Therefore, if the task states the email is to be written to my boss then preferences that were learned from emails to my boss will be retrieved.
>
> (2) We find that PROSE performs better by 33% due to our algorithmic contributions (iterative refinement and preference verification) and improved prompting. We find the biggest improvements relative to CIPHER are after the first step suggesting PROSE is more data efficient (cf. Appendix C.4. Figure 6). Additionally, we adapted PRELUDE to be able to provide demonstrations for an algorithm to learn from.
>
> We also improved PLUME’s ground truth preferences compared to PRELUDE such that the proxy humans are sensitive to them both when “writing” and “evaluating”. Additionally, when selecting the preference sets, we validated that they are not default behaviors in the LLMs, such as preferences like “brief” and “use bullet points” in PRELUDE. Having preferences that are default behaviors limits the utility of generation quality metrics for evaluating an algorithm’s ability to personalization as no personalization is needed.  When combined with our metric improvements, PLUME provides more meaningful results and evaluations.
>
> **Response to missing related work**
>
> Thank you for the feedback that more of a discussion of model alignment methods in the space of inference time interventions. We will make sure to expand our related work section to include these.

---

> > ### Comment · Reviewer_wsxk · 2025-04-08
> >
> > I have raised the score based on your response.

---

### Decision · Program_Chairs · 2025-05-01

**Decision:**

Accept (poster)

**Comment:**

This paper provides an improvement over a paper by Gao et al. 1] Gao, Ge, et al. "Aligning llm agents by learning latent preference from user edits." arXiv preprint arXiv:2404.15269 (2024).  To quote the reviewrs: Overall the paper is well-motivated and well-written. I believe the problem is an important one, the evaluation thorough, and this paper not only improves upon previous methods but also provides a better benchmark than PRELUDE. Graphs are nice and clear. The discussion on limitation addresses questions about computational costs.

There is no reason not to accept this paper - there are no major flaws and it is a solid contribution advancing the state of the art.  All reviewers agree with accept, the only question is on impact and importance.  The contribution is seen as incremental.

Here are the strengths of the paper as contributed by the reviewers:

.	This paper directly points out the weakness of the prelude framework. I have tested the code base of prelude before, and discovered the similar problems (metric, editing process) in the prelude as the authors pointed out in the paper. Thus, I believe this work's improvement over prelude is important and makes sense.
.	The proposed methods to improve the prelude is effective and can be directly measured by the common evaluation methods.
.	The PROSE framework is efficient and effective compared to other alignment methods in literature, e.g, reward model training, fine-tuning.
•	PROSE produces more precise preference descriptions than CIPHER – This is supported by experiments on PLUME, showing a 33% improvement in preference alignment.
•	Iterative refinement improves alignment between LLM generations and user preferences – Ablation studies demonstrate that increasing the number of refinement steps improves performance by 14.8%.
•	Consistency verification enhances robustness – Though its effect is smaller (1.5–1.7% improvement), the verification step helps filter irrelevant or overfit preferences.
•	PROSE and ICL are complementary – Combining PROSE with ICL leads improvement.
•	improved dataset PLUME - The authors analyze PRELUDE’s limitations (e.g., weak correlation between inferred preferences and generation quality) and show that PLUME provides more consistent evaluation metrics.


The weaknesses of the paper focus on the impact of the contribution.  All reviewers agree that it is a contribution, the debate is around how much of a contribution it is:

"The improvement over prelude basically focuses on introducing better evaluation metric, better process and prompting for editing, with additional verification process. However, there is no fundamental change compared to prelude. This makes the work seems to have limited contribution."

 "I would suggest use more complicated dataset, where the retrieval is not only based on the similarity between questions (or topic), but also based on occasions."

"This work seems to claim the refinement procedure is novel (From line 106 to the end of the paragraph). In fact, the prelude framework also includes the refinement of inferred preferences. If you check the code of prelude, it includes a step to aggregate the currently inferred preference with the previous ones."

This paper received three reviews all of which indicate "weak accept."  Despite a lack of discussion, all three reviewers acknowledged the rebuttal and two responded.  Two reviewers responded with one sentence comments.  One responded that they would raise their score in response to the authors rebuttal from weak reject to weak accept and one responded that they would maintain their score after reading the authors rebuttal.